# Regime Switching Bandits

Xiang Zhou [*†]   Yi Xiong [‡]   Ningyuan Chen [§]   Xuefeng Gao[¶]

## Abstract

We study a multi-armed bandit problem where the rewards exhibit regime switching. Specifically, the distributions of the random rewards generated from all arms are modulated by a common underlying state modeled as a finite-state Markov chain. The agent does not observe the underlying state and has to learn the transition matrix and the reward distributions. We propose a learning algorithm for this problem, building on spectral method-of-moments estimations for hidden Markov models, belief error control in partially observable Markov decision processes and upper-confidence-bound methods for online learning. We also establish an upper bound $O(T^{2/3}\sqrt{\log T})$ for the proposed learning algorithm where $T$ is the learning horizon. Finally, we conduct proof-of-concept experiments to illustrate the performance of the learning algorithm.

## 1   Introduction

The multi-armed bandit (MAB) problem is a popular model for sequential decision making with unknown information: the decision maker makes decisions repeatedly among $I$ different options, or arms. After each decision she receives a random reward having an *unknown* probability distribution that depends on the chosen arm. The objective is to maximize the expected total reward over a finite horizon of $T$ periods. The MAB problem has been extensively studied in various fields and applications including Internet advertising, dynamic pricing, recommender systems, clinical trials and medicine [12, 13, 43]. In the classical MAB problem, it is typically assumed that the random reward of each arm is i.i.d. (independently and identically distributed) over time and independent of the rewards from other arms. However, these assumptions do not necessarily hold in practice [10]. To address the drawback, a growing body of literature studies MAB problems with non-stationary rewards to capture temporal changes in the reward distributions in applications, see e.g. [11, 16, 20].

In this paper, we study a non-stationary MAB model with Markovian regime-switching rewards. We assume that the random rewards associated with all the arms are modulated by a *common unobserved* state (or regime) $\{M_t : t = 1, 2, \ldots\}$ modeled as a finite-state discrete-time Markov chain. This chain makes a transition at each period regardless of which arm is pulled and its transition probabilities are independent of the action chosen. Given $M_t = m$, the reward of arm $i$ is i.i.d., whose distribution is denoted $Q(\cdot|m, i)$. Such structural change of the environment is usually referred to as regime switching in finance [33]. The agent doesn't observe or control the underlying state $M_t$, and has to learn the transition probability matrix $P$ of $\{M_t\}$ as well as the distribution of reward of each arm $Q(\cdot|m, i)$, based on the observed historical rewards. The goal of the agent is to design a learning policy that decides which arm to pull in each period to minimize the expected regret over $T$ periods.

---
[*]The first two authors (Xiang Zhou and Yi Xiong) have equal contribution.

[†]Department of Systems Engineering and Engineering Management, The Chinese University of Hong Kong; 1911606962@qq.com

[‡]Department of Systems Engineering and Engineering Management, The Chinese University of Hong Kong; yxiong@se.cuhk.edu.hk

[§]The Rotman School of Management, University of Toronto; ningyuan.chen@utoronto.ca

[¶]Department of Systems Engineering and Engineering Management, The Chinese University of Hong Kong; xfgao@se.cuhk.edu.hk

35th Conference on Neural Information Processing Systems (NeurIPS 2021).

The regime-switching models are widely used in various industries. For example, in finance, the sudden changes of market environments are usually modeled as hidden Markov chains. In revenue management and marketing, a firm may face a shift in consumer demand due to undetected changes in sentiment or competition. In such cases, when the agents (traders or firms) take actions (trading financial assets with different strategies or setting prices), they need to learn the reward and the underlying state at the same time. Our setup is designed to tackle such problems.

**Our Contribution.** Our study features novel designs in three regards: we propose a new bandit problem formulation, develop a new learning algorithm, and prove regret bounds with new techniques.

In terms of problem formulation, online learning with unobserved states has attracted some attention recently [8, 19]. We consider the strongest oracle among the studies, who knows $P$ and $Q(\cdot|m,i)$, but doesn't observe the hidden state $M_t$. The oracle thus faces a partially observable Markov decision process (POMDP) [26] (with a long-run average reward objective). By reformulating it as a Markov decision process (MDP) with a continuous belief space (i.e., a distribution over hidden states), the oracle then solves the optimal policy (mapping belief states to actions) using the Bellman equation. Having sublinear regret benchmarked against the strong oracle, our algorithm has better theoretical performance than others with weaker oracles such as the best fixed arm [19] or memoryless policies (the action only depends on the current observation) [8].

In terms of algorithmic design, we propose a learning algorithm (see Algorithm 2) with two key ingredients. First, it builds on the recent advance on the estimation of the parameters of hidden Markov models (HMMs) using spectral method-of-moments methods [3, 2, 8]. It benefits from the theoretical finite-sample bound of spectral estimators, while the finite-sample guarantees of other alternatives such as maximum likelihood estimators remain an open problem [30]. Second, it builds on the well-known "upper confidence bound" (UCB) method in reinforcement learning [7, 25]. There are two difficulties here as the oracle uses the optimal (belief-based) policy of the POMDP. First, the spectral method can not use the non i.i.d. samples generated from the belief-based policy due to the complex history dependency. Second, the belief of the hidden state is subject to the estimation error. Hence, we divide the horizon into nested exploration and exploitation phases. We use spectral estimators in the exploration phase to gauge the estimation error of $P$ and $Q(\cdot|m,i)$. We use the UCB method to control the regret in the exploitation phase. Different from other learning problems, we re-calibrate the belief at the beginning of each exploitation phase based on the parameters estimated in the most recent exploration phase using previous exploratory samples.

In terms of technical analysis, we establish a regret bound of $O(T^{2/3}\sqrt{\log(T)})$ for our proposed learning algorithm where $T$ is the learning horizon. Our regret analysis draws inspirations from [25, 35] for learning MDPs and undiscounted reinforcement learning problems, but the analysis differs significantly from theirs since there are two main technical challenges in our problem.

First, to control the regret, we need to control of the error of the belief state, which itself is not directly observed and needs to be estimated. This is in stark contrast to learning MDPs [25, 35] with observed states. Specifically, we need to bound the estimation error of belief states by the estimation errors of the model parameters. In addition, since the belief state is un-observable, we also need to bound the error in the belief transition kernel, which measures the distance between the belief transition kernel under the optimistic belief MDP model (from the UCB component of our algorithm) at each episode and the belief transition kernel under the true model. These bounds are not trivial since the transition kernel of the belief state depends on the model parameters in a complex way via Bayesian updating. We overcome the difficulties by building on [18] and a delicate analysis of the belief transition kernel to control the errors in the estimations of belief states and the belief transitions.

Second, to establish regret bound, we need an explicit bound for the span of the bias function (also referred as the relative value function) for the belief MDP which has a continuous state space. Such a bound is often critical in the regret analysis of undiscounted reinforcement learning of continuous MDP, but it is either *taken as an assumption* [39] or proved under Hölder continuity assumptions that do not hold for the belief transitions in our setting [35, 27]. We overcome this challenge and bound the bias span by developing a novel approach, which could be of independent interest for learning continuous-state MDPs. Specifically, we bound the bias span by bounding the Lipschitz module of the bias function for our infinite-horizon undiscounted problem (with a long-run average reward objective). To achieve this, we rely on a non-trivial application of the results in [23] which provide general tools for proving Lipschitz continuity of value functions in *finite-horizon discounted* MDPs. One key step is to bound the Lipschitz module of the belief transitions using the Kantorovich

metric, which allows us to establish a bound on the Lipschitz module of the value function of the finite-horizon discounted problem *uniformly over the discounting factors*. Exploiting the connection with the infinite-horizon undiscounted problem then yields an explicit bound on the bias span for our problem. We also mention that our bound on the bias span is unrelated to the diameter of the POMDP discussed in [8]. The diameter in [8] is only for observation-based policies, not for belief-state based policies we consider. That also explains why we need a new approach to bound the bias span.

**Related Work.** Studies on non-stationary/switching MAB investigate the problem when the rewards of all arms may change over time [5, 20, 10, 6]. Our formulation can be regarded as non-stationary MAB with a special structure. They consider an even stronger oracle than ours, the best arm in each period. However, the total number of changes or the changing budget have to be sublinear in $T$ to achieve sublinear regret. In our formulation, the total number of transitions of the underlying state is linear in $T$, and the algorithms proposed in these papers fail even considering our oracle (See Section 5). Other studies focus on linear changing budget with some structure such as seasonality [15], which is not present in this paper.

Our work is also related to the restless Markov bandit problem [21, 36, 44] in which the state of each arm evolves according to independent Markov chains. In contrast, our regime-switching MAB model assumes a *common* underlying Markov chain so that the rewards of all arms are correlated, and the underlying state is unobservable to the agent. In addition, our work is related to MAB studies where rewards of all arms depend on a common unknown parameter or a latent random variable, see, e.g., [4, 22, 28, 32, 34]. Our model differs from them in that the common latent state variable follows a dynamic stochastic process which introduces difficulties in algorithm design and analysis.

Two papers have similar settings to ours. [19] studies MAB problems whose rewards are modulated by an unobserved Markov chain and the transition matrix may depend on the action. However, their oracle is the best fixed arm when defining the regret, which is much weaker than the optimal policy of the POMDP (the performance gap is linear between the two oracles). Therefore, their algorithm is expected to have linear regret when using our oracle. [8] proposes an algorithm based on spectral estimators for learning POMDPs. Their algorithm will generally suffer linear regret in our problem setting. This is because we consider a stronger oracle which is the optimal POMDP policy, whereas their oracle is the optimal memoryless policy, i.e., a policy that only depends on the current reward observation instead of using all historical observations to form the belief of the underlying state. It is known that memoryless policies are in general not optimal and hence the gap between their oracle and our oracle can be linear. Note that considering the memoryless policy allows [8] to circumvent the introduction of the belief entirely. Indeed, the dynamics under the memoryless policy can be viewed as a finite-state (modified) HMM and spectral estimators can be applied. Instead, because we consider the optimal belief-based policy, such reduction is not available. Our algorithm and regret analysis hinge on the interaction between the estimation of the belief and the spectral estimators. Our algorithm needs separate exploration to apply the spectral methods and uses the belief-based policy for exploitation. For the regret analysis, unlike [8], we have to carefully control the belief error and bound the span of the bias function from the optimistic belief MDP in each episode. The comparison of our study with related papers is summarized in Table 1. Note that we only present the regret in terms of $T$.

| Papers | Oracle | Changing Budget | Regret |
|--------|--------|-----------------|--------|
| [5] | Best fixed action | Linear | $\tilde{O}(\sqrt{T})$ |
| [20] | Best action in each period | Finite | $\tilde{O}(\sqrt{T})$ |
| [10] | Best action in each period | Sublinear | $\tilde{O}(T^{2/3})$ |
| [8] | Optimal memoryless policy | Linear | $\tilde{O}(\sqrt{T})$ |
| This paper | Optimal POMDP policy | Linear | $\tilde{O}(T^{2/3})$ |

Table 1: Comparison of our study with some related literature.

## 2 MAB with Markovian Regime Switching

### 2.1 Problem Formulation

Consider the MAB problem with arms $\mathcal{I} := \{1, \dots, I\}$. There is a Markov chain $\{M_t\}$ with states $\mathcal{M} := \{1, 2, \dots, M\}$ and transition probability matrix $P \in \mathbb{R}^{M \times M}$. In period $t = 1, 2, \dots$, if the state of the Markov chain $M_t = m$ and the agent chooses arm $I_t = i$, then the reward in that period

is $R_t$ with discrete finite support, and its distribution is denoted by $Q(\cdot|m,i) := \mathbb{P}(R_t \in \cdot|M_t = m, I_t = i)$, with $\mu_{m,i} := \mathbb{E}[R_t|M_t = m, I_t = i]$. We use $\boldsymbol{\mu} := (\mu_{m,i}) \in \mathbb{R}^{M \times I}$ to denote the mean reward matrix. The agent knows $M$ and $I$, but has no knowledge about the underlying state $M_t$ (also referred to as the regime), the transition matrix $P$ or the reward distribution $Q(\cdot|m,i)$. The goal is to design a learning policy that is adapted to the filtration generated by the observed rewards to decide which arm to pull in each period to maximize the expected cumulative reward over $T$ periods where $T$ is unknown in advance.

If an oracle knows $P$, $Q(\cdot|m,i)$ and the underlying state $M_t$, then the problem becomes trivial as s/he would select $I_t^* = \operatorname{argmax}_{i \in \mathcal{I}} \mu_{M_t,i}$ in period $t$. If we benchmark a learning policy against the oracle, then the regret must be linear in $T$, because the oracle always observes $M_t$ while the agent cannot predict the transition based on the history. Whenever a transition occurs, there is non-vanishing regret incurred. Since the number of transitions during $[0,T]$ is linear in $T$, the total regret is of the same order. Since comparing to the oracle knowing $M_t$ is uninformative, we consider a weaker oracle who knows $P$, $Q(\cdot|m,i)$, but not $M_t$. In this case, the oracle solves a POMDP since the states $M_t$ are unobservable and the optimal policy maps belief states (a distribution over the hidden state) to actions. The total expected reward of the POMDP scales linearly in $T$, and asymptotically the reward per period converges to a constant denoted by $\rho^*$ under the optimal belief-based policy. See Section 2.2.

For a learning policy $\pi$, we denote by $R_t^\pi$ the reward received under the learning policy $\pi$ (which does not know $P$, $Q(\cdot|m,i)$ initially) in period $t$. We follow the literature (see, e.g., [25, 36, 1]) and define its total regret after $T$ periods by

$$\mathcal{R}_T := T\rho^* - \sum_{t=1}^{T} R_t^\pi. \tag{1}$$

Our goal is to design a learning algorithm with theoretical guarantees including high probability and expectation bounds (sublinear in $T$) on the total regret.

Without loss of generality we consider Bernoulli rewards with mean $\mu_{m,i} \in (0,1)$ for all $m, i$. Hence $\mu_{m,i}$ characterizes the distribution $Q(\cdot|m,i)$. Our analysis holds generally for random rewards with discrete finite support. In addition, we impose the following assumptions.

**Assumption 1.** *The transition matrix $P$ of the Markov chain $\{M_t\}$ is invertible.*

**Assumption 2.** *The mean reward matrix $\boldsymbol{\mu} = (\mu_{m,i})$ has full row rank.*

**Assumption 3.** *The smallest element of the transition matrix $\epsilon := \min_{i,j \in \mathcal{M}} P_{ij} > 0$.*

The first two assumptions are required for finite-sample guarantees of spectral estimators for HMMs [3, 2]. The third assumption is needed to control the belief error of the hidden states by the state-of-art methods, see [18] and our Proposition 3. We next reformulate the POMDP as a belief MDP.

## 2.2 Reduction of POMDP to Belief MDP

To present our learning algorithm and analyze the regret, we first investigate the POMDP problem faced by the oracle where parameters $P$, $\boldsymbol{\mu}$ (equivalently $Q$ for Bernoulli rewards) are known with unobserved states $M_t$. Based on the historical observed history, the oracle forms a belief of the underlying state. The belief can be encoded by a $M$-dimension vector $b_t = (b_t(1), \ldots, b_t(M)) \in \mathcal{B}$: $b_t(m) := \mathbb{P}(M_t = m|I_1, \cdots, I_{t-1}, R_1, \cdots, R_{t-1})$, where $\mathcal{B} := \left\{ b \in \mathbb{R}_+^M : \sum_{m=1}^{M} b(m) = 1 \right\}$. It is well known that the POMDP of the oracle can be seen as a MDP built on a (continuous) belief state space $\mathcal{B}$, and here a policy is a mapping from belief states to actions, see [26].

We next introduce a few notations that facilitate the analysis. For notation simplicity, we write $c(m,i) := \mu_{m,i}$. Given belief $b \in \mathcal{B}$, the expected reward of arm $i$ is $\bar{c}(b,i) := \sum_{m=1}^{M} c(m,i)b(m)$. In period $t$, given belief $b_t = b$, $I_t = i$, $R_t = r$, by Bayes' theorem, the belief $b_{t+1}$ is updated by

$b_{t+1} = H_{\boldsymbol{\mu},P}(b,i,r)$, where the $m$-th entry is $b_{t+1}(m) = \dfrac{\sum\limits_{m'} P(m',m) \cdot (\boldsymbol{\mu}_{m',i})^r (1-\boldsymbol{\mu}_{m',i})^{1-r} \cdot b_t(m')}{\sum\limits_{m''} (\boldsymbol{\mu}_{m'',i})^r (1-\boldsymbol{\mu}_{m'',i})^{1-r} \cdot b_t(m'')}$. It

is obvious that the forward function $H$ depends on the transition matrix $P$ and the reward matrix $\boldsymbol{\mu}$. We can also define the transition probability of the belief state conditional on the arm pulled: $\bar{T}(\cdot|b,i) := \mathbb{P}(b_{t+1} \in \cdot|b,i)$, where $b_{t+1}$ is random due to the random reward.

The long-run average reward of the infinite-horizon belief MDP following policy $\pi$ given the initial belief $b$ can be written as $\rho_b^\pi := \limsup_{T\to\infty} \frac{1}{T}\mathbb{E}[\sum_{t=1}^T R_t^\pi | b_1 = b]$. The optimal (belief-based) policy maximizes $\rho_b^\pi$ for a given $b$. One can show that $\sup_\pi \rho_b^\pi$ is independent of the initial belief $b$ (Proposition 8.2.1 [38]). Therefore, we use $\rho^* := \sup_\pi \rho_b^\pi$ to denote the optimal long-run average reward. Under this belief MDP formulation, for all $b \in \mathcal{B}$, the Bellman equation states that

$$\rho^* + v(b) = \max_{i\in\mathcal{I}} \left[ \bar{c}(b,i) + \int_{\mathcal{B}} \bar{T}(db'|b,i)v(b') \right], \tag{2}$$

where $v : \mathcal{B} \mapsto \mathbb{R}$ is the bias function. It can be shown (see the proof of Proposition 2 in Appendix) that under our assumptions, $\rho^*$ and $v(b)$ are well defined and there exists a stationary deterministic optimal policy $\pi^*$ which maps a belief state to an arm to pull (an action that maximizes the right side of (2)). Finding the optimal policy for the POMDP model is computationally intractable in general [37, 31]. Therefore, various approximate methods have been proposed to solve (2) and to find the optimal policy for the belief MDP, see e.g. [46, 41, 42]. In this work, we do not focus on this planning problem for a known model, and we assume the access to an optimization oracle that solve (2) and returns the optimal average reward $\rho^*$ and the optimal stationary policy for a given known model.

## 3 The SEEU Algorithm

This section describes our learning algorithm for the regime switching MAB model: the Spectral Exploration and Exploitation with UCB (SEEU) algorithm. To device a learning policy for the POMDP with unknown $\boldsymbol{\mu}$ and $P$, one needs a procedure to estimate those quantities from observed rewards. [3, 2] propose the so-called spectral estimator for the unknown parameters in HMMs. However, the algorithm is not directly applicable to ours, because there is no decision making involved in HMMs . To use the spectral estimator, we divide the learning horizon $T$ into nested "exploration" and "exploitation" phases. In the exploration phase, we randomly select an arm in each period. This transforms the system into a HMM so that we can apply the spectral method to estimate $\boldsymbol{\mu}$ and $P$ from the observed rewards in the phase. In the exploitation phase, based on the estimators obtained from the exploration phase, we use a UCB-type policy to further narrow down the optimal belief-based policy in the POMDP introduced in Section 2.2.

### 3.1 Spectral Estimator

We introduce the spectral estimator [3, 2], and adapt it to our setting. To simplify the notation, suppose the exploration phase starts from period 1 until period $n$, with realized arms $\{i_1, \ldots, i_n\}$, and realized rewards $\{r_1, \ldots, r_n\}$ sampled from Bernoulli distributions. Recall $I$ is the cardinality of the arm set $\mathcal{I}$, then one can create a one-to-one mapping from a pair $(i, r)$ into a scalar $s \in \{1, 2, ..., 2I\}$. Therefore, the pair can be expressed as a vector $y \in \{0, 1\}^{2I}$ such that in each period $t$, $y_t$ satisfies $\mathbb{1}_{\{y_t = e_s\}} = \mathbb{1}_{\{r_t = r, i_t = i\}}$, where $e_s$ is a basis vector with its $s$-th element being one and zero otherwise. Let $A \in \mathbb{R}^{2I \times M}$ be the observation probability matrix conditional on the state: $A(s, m) = \mathbb{P}(R_t = r, I_t = i | M_t = m)$. It can be shown that $A$ satisfies $\mathbb{E}[y_t | M_t = m] = Ae_m$, and $\mathbb{E}[y_{t+1} | M_t = m] = AP^T e_m$. Write $\otimes$ for the tensor product. For three consecutive observations $y_{t-1}, y_t, y_{t+1}$, define

$$\widetilde{y}_{t-1} := \mathbb{E}[y_{t+1} \otimes y_t]\mathbb{E}[y_{t-1} \otimes y_t]^{-1}y_{t-1}, \qquad \widetilde{y}_t := \mathbb{E}[y_{t+1} \otimes y_{t-1}]\mathbb{E}[y_t \otimes y_{t-1}]^{-1}y_t,$$
$$M_2 := \mathbb{E}[\widetilde{y}_{t-1} \otimes \widetilde{y}_t], \qquad\qquad M_3 := \mathbb{E}[\widetilde{y}_{t-1} \otimes \widetilde{y}_t \otimes y_{t+1}].$$

From the observations $\{y_1, y_2, \ldots, y_n\}$, we may construct the estimations $\hat{M}_2$ and $\hat{M}_3$ for $M_2$ and $M_3$ respectively, and apply the tensor decomposition to obtain the estimator $\hat{\boldsymbol{\mu}}$ for the unknown mean reward matrix and $\hat{P}$ for the transition matrix. This procedure is summarized in Algorithm 1. We use $\hat{A}_m$ (respectively $\hat{B}_m$) to denote the $m$-th column vector of $\hat{A}$ (respectively $\hat{B}$). In addition, we have the following result ([8]) and it provides the confidence regions of the estimators in Algorithm 1.

**Proposition 1.** *Under Assumptions 1 and 2, for any $\delta \in (0, 1)$ and any initial distribution, there exists $N_0$ such that when $n \geq N_0$, with probability $1 - \delta$, the estimated $\hat{\boldsymbol{\mu}}$ and $\hat{P}$ by Algorithm 1*

**Algorithm 1** Spectral estimation of $(\boldsymbol{\mu}, P)$ from the observations from the exploration phase [2, 3, 8].

---

**Input:** sample size $n$, $\{y_1, y_2, \ldots, y_n\}$ created from the rewards $\{r_1, \ldots, r_n\}$ and arms $\{i_1, \ldots, i_n\}$
**Output:** The estimation $\hat{\boldsymbol{\mu}}, \hat{P}$
 1: For $i, j \in \{-1, 0, 1\}$: compute $\hat{W}_{i,j} = \frac{1}{N-2} \sum_{t=2}^{N-1} y_{t+i} \otimes y_{t+j}$.
 2: For $t = 2, \ldots, n-1$: compute $\hat{y}_{t-1} := \hat{W}_{1,0}(\hat{W}_{-1,0})^{-1} y_{t-1}$, $\hat{y}_t := \hat{W}_{1,-1}(\hat{W}_{0,-1})^{-1} y_t$.
 3: Compute $\hat{M}_2 := \frac{1}{N-2} \sum_{t=2}^{N-1} \hat{y}_{t-1} \otimes \hat{y}_t$, $\hat{M}_3 := \frac{1}{N-2} \sum_{t=2}^{N-1} \hat{y}_{t-1} \otimes \hat{y}_t \otimes y_{t+1}$.
 4: Apply tensor decomposition ([2]):
    $\hat{B} = \textbf{TensorDecomposition}(\hat{M}_2, \hat{M}_3)$.
 5: Compute $\hat{A}_m = \hat{W}_{-1,0}(\hat{W}_{1,0})^{\dagger} \hat{B}_m$ for each $m \in \mathcal{M}$.
 6: Return $m$th row vector $(\hat{\boldsymbol{\mu}})^m$ of $\hat{\boldsymbol{\mu}}$ from $\hat{A}_m$.
 7: Return $\hat{P} = (\hat{A}^{\dagger} \hat{B})^{\top}$ († represents the pseudoinverse of a matrix)

---

*satisfy*

$$||(\boldsymbol{\mu})^m - (\hat{\boldsymbol{\mu}})^m||_2 \leq C_1 \sqrt{\frac{\log(6\frac{S^2+S}{\delta})}{n}}, \quad m \in \mathcal{M},$$

$$||P - \hat{P}||_2 \leq C_2 \sqrt{\frac{\log(6\frac{S^2+S}{\delta})}{n}}. \tag{3}$$

*where $(\boldsymbol{\mu})^m$ and $(\hat{\boldsymbol{\mu}})^m$ are the $m$-th row vectors of $\boldsymbol{\mu}$ and $\hat{\boldsymbol{\mu}}$, respectively. Here, $S = 2I$, and $C_1$, $C_2$ are constants independent of $n$.*

The expressions of constants $N_0, C_1, C_2$ are given in Section B in the appendix. Note that parameters $\boldsymbol{\mu}, P$ are identifiable up to permutations of the hidden states [8].

## 3.2 The SEEU Algorithm

The SEEU algorithm proceeds in episodes of increasing length. As mentioned before, each episode is divided into exploration and exploitation phases. In episode $k$, it starts with the exploration phase that lasts for a fixed number of periods $\tau_1$, and the algorithm uniformly randomly chooses an arm and observes the rewards. After the exploration phase, the algorithm applies Algorithm 1 to (re-)estimate $\boldsymbol{\mu}$ and $P$. Moreover, it constructs a confidence interval based on Proposition 1 with a confidence level $1 - \delta_k$, where $\delta_k := \delta/k^3$ is a vanishing sequence. Then the algorithm enters the exploitation phase. Its length is proportional to $\sqrt{k}$. In the exploitation phase, it conducts UCB-type learning: the arm is pulled according to a policy that corresponds to the optimistic estimator of $\boldsymbol{\mu}$ and $P$ inside the confidence interval. The detailed steps are listed in Algorithm 2. Note we use the UCB component in the exploitation phase instead of point estimators of $\boldsymbol{\mu}$ and $P$ for the ease of analysis.

## 3.3 Discussions on the SEEU Algorithm

**Computations.** For given parameters $(\boldsymbol{\mu}, P)$, we need to compute the optimal average reward $\rho^*(\boldsymbol{\mu}, P)$ that depends on the parameters (Step 8 in Algorithm 2). Various computational and approximation methods have been proposed to tackle this planning problem for belief MDPs as mentioned in Section 2.2. In addition, we need to find out the optimistic POMDP in the confidence region $\mathcal{C}_k(\delta_k)$ with the best average reward. In general it is not clear whether there is an efficient computational method to find the optimistic plausible POMDP model in the confidence region when the unknown parameters are high-dimensional. The extended value iteration method in [25] does not work in our POMDP setting. This is because we cannot separately find $\boldsymbol{\mu}$ and $P$ as $\bar{c}$ and $\bar{T}$ in (2) both depend on $\boldsymbol{\mu}$, and we cannot write the inner optimization in the extended value iteration to find optimistic $P$ as a linear programming over the convex polytope $\mathcal{C}_k(\delta_k)$ since $\bar{T}$ is a nonlinear function of $\boldsymbol{\mu}$ and $P$. This issue is also present in recent studies on learning continuous-state MDPs with the upper confidence bound approach, see e.g. [27] for further discussions. For low dimensional models, one can discretize $\mathcal{C}_k(\delta_k)$ into grids and calculate the corresponding optimal average reward $\rho^*$ at each grid point so as to find (approximately) the optimistic model $(\boldsymbol{\mu}_k, P_k)$. The discretization error does affect the regret, although it can be controlled arbitrarily well with sufficient computational capacity.

---

**Algorithm 2** The SEEU Algorithm

---

**Input:** Initial belief $b_1$, precision $\delta$, exploration parameter $\tau_1$, exploitation parameter $\tau_2$

1: **for** $k = 1, 2, 3, \ldots$ **do**
2:      Set the start time of episode $k$, $t_k := t$
3:      **for** $t = t_k, t_k + 1, \ldots, t_k + \tau_1$ **do**
4:          Uniformly randomly select an arm: $\mathbb{P}(I_t = i) = \frac{1}{I}$
5:      **end for**
6:      Input the realized actions and rewards in all previous exploration phases $\hat{\mathcal{I}}_k :=$ $\{i_{t_1:t_1+\tau_1}, \cdots, i_{t_k:t_k+\tau_1}\}$ and $\hat{\mathcal{R}}_k := \{r_{t_1:t_1+\tau_1}, \cdots, r_{t_k:t_k+\tau_1}\}$ to Algorithm 1 to compute $\hat{\boldsymbol{\mu}}_k, \hat{P}_k = \textbf{Spectral Estimation}(\hat{\mathcal{I}}_k, \hat{\mathcal{R}}_k)$
7:      Compute the confidence interval $\mathcal{C}_k(\delta_k)$ from (3) using the confidence level $1 - \delta_k = 1 - \frac{\delta}{k^3}$ such that $\mathbb{P}\{(\boldsymbol{\mu}, P) \in \mathcal{C}_k(\delta_k)\} \geq 1 - \delta_k$
8:      Find the optimistic POMDP in the confidence interval
$$(\boldsymbol{\mu}_k, P_k) = \operatorname{argmax}_{(\boldsymbol{\mu}, P) \in \mathcal{C}(\delta_k)} \rho^*(\boldsymbol{\mu}, P)$$
9:      **for** $t = 1, 2, \ldots, t_k + \tau_1$ **do**
10:          Update belief $b_t^k$ to $b_{t+1}^k = H_{\boldsymbol{\mu}_k, P_k}(b_t^k, i_t, r_t)$ under the new parameters $(\boldsymbol{\mu}_k, P_k)$
11:      **end for**
12:      **for** $t = t_k + \tau_1 + 1, \ldots, t_k + \tau_1 + \tau_2\sqrt{k}$ **do**
13:          Execute the optimal policy $\pi^{(k)}$ by solving the Bellman equation (2) under parameters $(\boldsymbol{\mu}_k, P_k)$: $i_t = \pi^{(k)}(b_t^k)$
14:          Observe reward $r_t$
15:          Update the belief at $t + 1$ by $b_{t+1}^k = H_{\boldsymbol{\mu}_k, P_k}(b_t^k, i_t, r_t)$
16:      **end for**
17: **end for**

---

Below we discuss its impact on regret. The discretization kicks in when we want to find the optimistic POMDP in the confidence region, whose gain is denoted as $\rho^k$ in episode $k$. In practice, we have to discretize the confidence region into grid points and find the optimistic POMDP among the finite set. Suppose one can obtain an approximate optimistic model with error $\epsilon_k$, that is, suppose we can find a model with the gain $\tilde{\rho}_k \geq \rho^k - \epsilon_k$. Then we can infer from formula (22) in the appendix that the extra regret incurred due to the discretization error is given by $\sum_{k=1}^{K} \sum_{t \in E_k} \epsilon_k = \tau_2 \sum_{k=1}^{K} \sqrt{k}\epsilon_k$, where $E_k$ denotes the exploitation phase in episode $k$. One can show that the order of $K$, the number of episodes up to time $T$, is $T^{2/3}$. Hence if the discretization error can be controlled at $\epsilon_k = c/\sqrt{k}$, then the extra regret is simply $c\tau_2 K$, which is $O(T^{2/3})$. On the other hand, if the discretization error $\epsilon_k$ is a constant $c > 0$ in all the exploitation phases, then the extra regret incurred is of the order $\sum_{k=1}^{K} \sqrt{k}c \approx c \cdot K^{3/2}$, which is of order $cT$. There is a trade-off between the additional computational complexity due to discretization and the regret bound. For instance, to control the discretization error at $\epsilon_k = c/\sqrt{k}$, the computational cost is higher compared with the case $\epsilon_k = c$. In general, it remains open to find efficient methods to solve the optimistic POMDP approximately in the high-dimensional setting. In our regret analysis below, we do not take into account approximation errors arising from the computational aspects discussed above, as in [35, 27].

**Dependence on the unknown parameters.** When computing the confidence region in Step 7 of Algorithm 2, the agent needs the information of the constants $C_1$ and $C_2$ in Proposition 1. These constants depend on a few "primitives" that can be hard to know, for example, the mixing rate of the underlying Markov chain. However we only need upper bounds for $C_1$ and $C_2$ for the theoretical guarantee, and hence a rough and conservative estimate would be sufficient. Such dependence on some unknown parameters is common in learning problems, and one remedy is to dedicate the beginning of the horizon to estimate the unknown parameters, which typically doesn't increase the rate of the regret. Alternatively, $C_1$ and $C_2$ can be replaced by parameters that are tuned by hand. See Remark 3 of [8] for a further discussion on this issue.

## 4 Regret Bound

This section presents our main theoretical results on the regret bound for the SEEU algorithm. We first state two technical results that are important in proving the regret bounds.

**Proposition 2** (Uniform bound on the bias span). *If the belief MDP satisfies Assumption 3, then for $(\rho, v)$ satisfying the Bellman equation* (2), *we have the span of the bias function* $\text{span}(v) := \max_{b \in \mathcal{B}} v(b) - \min_{b \in \mathcal{B}} v(b)$ *is bounded by* $D(\epsilon)$, *where*

$$D(\epsilon) := \frac{8 \left( \frac{2}{(1-\alpha)^2} + (1+\alpha) \log_\alpha \frac{1-\alpha}{8} \right)}{1 - \alpha}, \quad with \ \alpha = \frac{1 - 2\epsilon}{1 - \epsilon} \in (0, 1).$$

Recall $v_k$ is the bias function for the optimistic belief MDP in episode $k$. Proposition 2 guarantees that $\text{span}(v_k)$ is bounded by $D = D(\epsilon/2)$ uniformly in $k$, because Assumption 3 can be satisfied (with $\epsilon$ replaced by $\epsilon/2$) by the optimistic MDPs when $T$ is sufficiently large due to Proposition 1.

**Proposition 3** (Controlling the belief error). *Suppose Assumption 3 holds. Given* $(\hat{\boldsymbol{\mu}}, \hat{P})$, *an estimator of the true model parameters* $(\boldsymbol{\mu}, P)$. *For an arbitrary reward-action sequence* $\{r_{1:t}, i_{1:t}\}_{t \geq 1}$, *let* $\hat{b}_t$ *and* $b_t$ *be the corresponding beliefs in period* $t$ *under* $(\hat{\boldsymbol{\mu}}, \hat{P})$ *and* $(\boldsymbol{\mu}, P)$ *respectively. Then there exists constants* $L_1, L_2$ *such that*

$$||\hat{b}_t - b_t||_1 \leq L_1 ||\hat{\boldsymbol{\mu}} - \boldsymbol{\mu}||_1 + L_2 ||\hat{P} - P||_F, \tag{4}$$

*where* $L_1 = \frac{4M(1-\epsilon)^2}{\epsilon^2 \min\{\boldsymbol{\mu}_{\min}, 1 - \boldsymbol{\mu}_{\max}\}}$, $L_2 = \frac{4M(1-\epsilon)^2}{\epsilon^3} + \sqrt{M}$, $||\cdot||_F$ *is the Frobenius norm,* $\boldsymbol{\mu}_{\max}$ *and* $\boldsymbol{\mu}_{\min}$ *are the maximum and minimum element of the matrix* $\boldsymbol{\mu}$ *respectively.*

We now state our first main result. The proof is given in Appendix E.

**Theorem 1.** *Suppose Assumptions 1 to 3 hold. Fix the parameter* $\tau_1$ *in Algorithm 2 to be sufficiently large. Then there exist constants* $T_0, C$ *which are independent of* $T$, *such that when* $T > T_0$, *with probability* $1 - \frac{7}{2}\delta$, *the regret of Algorithm 2 satisfies*

$$\mathcal{R}_T \leq C T^{2/3} \sqrt{\log \left( \frac{9(S+1)}{\delta} T \right)} + T_0 \rho^*,$$

*where* $S = 2I$ *and* $\rho^*$ *denotes the optimal long-run average reward under the true model.*

The constant $T_0$ measures the number of periods needed for the sample size in the exploration phases to exceed $N_0$ arising in Proposition 1. The constant $C$ has the following expression:

$$C = 3\sqrt{2} \left[ \left( D + 1 + \left( 1 + \frac{D(1-\alpha)}{2} \right) L_1 \right) M^{3/2} C_1 + \left( 1 + \frac{D(1-\alpha)}{2} \right) L_2 M^{1/2} C_2 \right] \tau_2^{1/3} \tau_1^{-1/2}$$

$$+ 3\tau_2^{-2/3}(\tau_1 \rho^* + D) + (D+1)\sqrt{2 \ln(\frac{1}{\delta})}.$$

Here, $M$ is the number of hidden states, $C_1, C_2$ are given in Proposition 1, $\alpha, D$ is given in Proposition 2, and $L_1, L_2$ are given in Proposition 3. One can verify that $C = \tilde{O}(\epsilon^{-4} C_1 M^{5/2} + \epsilon^{-5} C_2 M^{3/2})$. Here we require $\tau_1$ to be large so as to apply Proposition 1 from the first episode to simplify the presentation. Theoretically, this requirement is not essential and can be removed without affecting the order of the regret. For $\tau_2$, it can be an arbitrary positive integer in our regret analysis.

**Remark 2** (Effect of lengths of exploration and exploitation phases on regret order). *In Algorithm 2, we use an exploration phase of length* $\tau_1$ *and a exploitation phase of length* $\tau_2\sqrt{k}$ *in episode* $k$. *This in the end leads to* $O(T^{2/3})$ *of the regret as the total length of exploration (and the total number of episodes) is* $\Theta(T^{2/3})$. *The intuition of such a choice is the following. After* $k$ *episodes, the total length of exploration is* $\tau_1 k$, *and hence the estimation error of the model parameters and also the belief states is of order* $1/\sqrt{k}$. *As a result, we needs to choose an exploitation phase with length* $\tau_2\sqrt{k}$ *to balance this so that the regret incurred per episode is controlled. One might wonder whether one can simply choose a longer exploitation phase of length* $\tau_2 k^\alpha$ *with* $\alpha \geq 1/2$, *and then the regret order will be reduced. This is not possible, and the intuition is as follows. The total number of episodes* $K$ *satisfies* $\sum_{k=1}^{K}(\tau_1 + \tau_2 k^\alpha) = T$, *which yields* $K \sim T^{\frac{1}{\alpha+1}}$. *The total regret incurred then is, ignoring the logarithmic factors, on the order of* $\sum_{k=1}^{K}(\tau_1 + \frac{1}{\sqrt{k}} \cdot \tau_2 k^\alpha)$, *which is* $T^{\frac{\alpha+1/2}{\alpha+1}}$. *One can immediately see that the choice of* $\alpha = 1/2$ *is actually optimal, leading to a regret of order* $T^{2/3}$. *The impact of the hyper-parameters* $\tau_1$ *and* $\tau_2$ *on the regret will be studied numerically in Section 5.*

From Theorem 1, we can choose an appropriate $\delta = \frac{9(S+1)}{T}$ and readily obtain the following expectation bound on the regret. The proof is omitted.

**Theorem 3.** *Under the same assumptions as Theorem 1, the regret of Algorithm 2 satisfies*

$$\mathbb{E}[\mathcal{R}_T] \leq CT^{2/3}\sqrt{2\log T} + (T_0 + 32(S+1))\rho^*.$$

**Remark 4** (Lower bound). *For the lower bound of the regret, consider the $I$ problem instances (equal to the number of arms): In instance $i$, let $\mu_{m,i} = 0.5 + \bar{\epsilon}$ for all $m$ for a small positive constant $\bar{\epsilon}$, and let $\mu_{m,j} = 0.5$ for all $m$ and $j \neq i$. Such structure makes sure that the oracle policy simply pulls one arm without the need to infer the state. Since the problem reduces to the classic MAB, the regret is at least $O(\sqrt{IT})$ in this case. Note that the setup of the instances may violate Assumption 2, but this can be easily fixed by introducing an arbitrarily small perturbation to $\boldsymbol{\mu}$. The gap between the upper and lower bounds is probably caused by the split of exploration/exploitation phases in our algorithm. In the exploration phase, arms have to be pulled purely randomly in order to estimate the parameters. Such naive exploration without any maximization may lead to unnecessary regret. In fact, it resembles the structure of the explore-then-commit algorithm (Chapter 6 of [29]) for the classic MAB problem, which turns out to have the suboptimal regret $O(T^{2/3})$ as well. With special structures, such as a common mean for all states, special-purpose algorithms can be designed to achieve the optimal rate $O(T^{1/2})$. Unfortunately, our algorithm doesn't adapt to the special structure. In fact, the spectral estimator no longer works because of the degeneracy as it cannot differentiate between states from the observations. One natural direction that may potentially improve our algorithm and close the gap is to integrate the exploration and exploitation. We may follow the design similar to UCRL2 in [25]: in each episode, we use the parameters estimated in previous episodes and use the optimistic policy in the confidence region. The observations are then used to update the estimation. This can lower the cost for exploration because the naive exploration is replaced by a near-optimal policy when the confidence region is small. However, it requires the spectral estimator to work with non i.i.d. samples generated by the optimistic belief-based policy. The design of the spectral estimator makes it hard to handle non i.i.d. observations with complex history dependency. This is also why [8] focused on observation-based policies in order to apply the spectral estimator. We may replace the spectral estimator by other estimators with better theoretical properties. The likelihood-based estimators may be a promising candidate [45], but they have not been fully extended to POMDPs yet. In summary, we need better estimators for POMDP/HMMs (which is a dynamic research area itself) with finite-sample guarantees to improve the upper bound. Nevertheless, we are not aware of other algorithms that can achieve sublinear regret in our setting.*

**Remark 5** (General reward distribution). *Our model requires the estimation of the reward distribution instead of just the mean to estimate the belief. For discrete rewards taking $O$ possible values, the regret bound holds with $S = OI$ instead of $2I$ for Bernoulli rewards. For continuous reward distributions, it might be possible to combine the non-parametric HMM inference method in [18] with our algorithm design to obtain regret bounds, and we leave it for future work.*

## 5 Numerical Experiment

In this section, we present proof-of-concept experiments. Note that large-scale POMDPs with long-run average objectives (the oracle in our problem) are computationally difficult to solve, i.e. it is challenging to find the optimal belief-based policy [14]. On the other hand, while there can be many hidden states in general, often only two or three states are important to model in several application areas, e.g. "bull market" and "bear market" in finance [17]. Hence we focus on small-scale experiments, following some recent literature on reinforcement learning for POMDPs [8, 24].

As a representative example, we consider a 2-hidden-state, 2-arm setting with $P = \begin{bmatrix} 1/3 & 2/3 \\ 3/4 & 1/4 \end{bmatrix}$ and $\boldsymbol{\mu} = \begin{bmatrix} 0.9 & 0.1 \\ 0.5 & 0.6 \end{bmatrix}$, where the random reward follows a Bernoulli distribution. We compare our algorithm with (1) $\epsilon$-greedy ($\epsilon = 0.1$), non-stationary bandits algorithms including (2) Sliding-Window UCB (SW-UCB) [20] with tuned window size, (3) Exp3.S [5] with $L = T$ (the hyperparameter $L$ is the number of changes in their algorithm), and (4) the optimal memoryless policy for POMDPs discussed in [8]. To implement (4) under our model setting, we assume the reward $\mu$ and the transition matrix $P$ are known and search for the optimal stochastic memoryless policy that maps the reward and action in the last period to a discrete distribution over two arms. We also run an algorithm/oracle

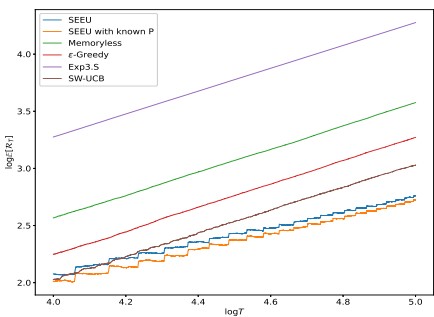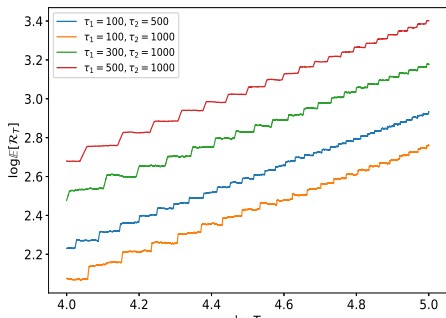

Figure 1: (a) Regret comparison of different algorithms; (b) Effect of $(\tau_1, \tau_2)$ on the regret of SEEU.

that implements SEEU with known $P$ but unknown $\mu$. This can allow us to better understand the value of knowing the information of $P$. In Figure 1(a), we plot the average regret versus $T$ of different algorithms in log-log scale, where the number of runs for each algorithm is 500. The numerical experiments are conducted on a PC with 3.10 GHz Intel Processor and 16 GB of RAM. We observe that the slopes of all algorithms except for two SEEU implementations are close to one, suggesting that they incur linear regrets. This is expected, because these algorithms don't take into account the hidden states. On the other hand, the slope of SEEU is close to $2/3$. This is consistent with our theoretical result (Theorem 3). Similar observations are made on other small-scale examples. This demonstrates the effectiveness of our SEEU algorithm, particularly when the horizon length $T$ is relatively large.

We also briefly discuss the impact of parameters $\tau_1$ and $\tau_2$ on the performance of the SEEU algorithm. For the example above, we calculate the average regret for several pairs of parameters $(\tau_1, \tau_2)$. It can be seen that the choices of these parameters do not affect the order $O(T^{2/3})$ of the regret (the slope). See Figure 1(b) for an illustration.

## 6    Conclusions, Limitations and Future Research

In this paper, we study a non-stationary MAB model with Markovian regime-switching rewards. We propose a learning algorithm that integrates spectral estimators for hidden Markov models and upper confidence methods from reinforcement learning. We also establish a regret bound of order of $O(T^{2/3}\sqrt{\log T})$ for the learning algorithm. As far as we know, this is the first algorithm with sublinear regret for MAB with unobservable regime switching.

The main limitation of our work is that the regret upper bound does not match the lower bound. It would be interesting to find out whether one can improve the regret upper bound $O(T^{2/3}\sqrt{\log T})$. The gap in the upper and lower bounds are likely to be filled if one can resolve either of the following two fundamental open questions: (1) show the spectral method can be applied to non i.i.d. samples generated from belief-based policies, and (2) establish finite-sample guarantees for other estimators (e.g. maximum likelihood estimators) of POMDP parameters with data generated from adaptive policies. We leave them for future research.

## Acknowledgments and Disclosure of Funding

The authors thank the Area Chair and three anonymous referees for constructive comments and helpful suggestions. Xuefeng Gao acknowledges support from Hong Kong RGC GRF Grants 14201520 and 14201421. The research of Ningyuan Chen is partially supported by NSERC Discovery Grants RGPIN-2020-04038.

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
