## A Table of Notations

| Notation | Description |
|----------|-------------|
| $T$ | The length of decision horizon |
| $P$ | The transition matrix of underlying states |
| $\boldsymbol{\mu}$ | The mean reward matrix |
| $\mathcal{M}$ | The set of underlying states |
| $\mathcal{I}$ | The set of arms |
| $\mathcal{R}$ | The set of rewards |
| $M_t$ | The underlying state at time $t$ |
| $I_t$ | The chosen arm at time $t$ |
| $R_t$ | The random reward at time $t$ |
| $\mathcal{F}_t$ | The history up to time $t$ |
| $\rho^*$ | The optimal long term average reward |
| $\mathcal{R}_T$ | Regret during the total horizon |
| $\boldsymbol{b_t}$ | The belief state at time $t$ |
| $\mathcal{B}$ | The set of belief states |
| $c(m, i)$ | The reward function given the state and arm |
| $\bar{c}(b, i)$ | The reward function w.r.t. belief state |
| $Q$ | The reward distribution |
| $H$ | The belief state forward kernel |
| $\bar{T}$ | The transition kernel w.r.t. belief state |
| $D$ | The uniform upper bound of $\mathrm{span}(v)$ |
| $\mathbb{E}^\pi$ | Taken expectation respect to the true parameters $\boldsymbol{\mu}$ and $P$ under policy $\pi$ |
| $\mathbb{E}^\pi_k$ | Taken expectation respect to estimated parameters $\boldsymbol{\mu}_k$ and $P_k$ under policy $\pi$ |

Table 2: Summary of notations

## B Constants in Proposition 1

We consider the action-reward pair $(R_t, I_t)$ as our observation of the underlying state $M_t$. We encode the pair $(r, i)$ into a variable $s \in \{1, 2, ..., 2I\}$ through a suitable one-to-one mapping. We rewrite our observable random vector $(R_t, I_t)$ as a random variable $S_t$. Hence we can define the following matrix $A_1, A_2, A_3 \in \mathbb{R}^{2I \times M}$, where

$$
\begin{aligned}
A_1(s, m) &= \mathbb{P}(S_{t-1} = s | M_t = m), \\
A_2(s, m) &= \mathbb{P}(S_t = s | M_t = m), \\
A_3(s, m) &= \mathbb{P}(S_{t+1} = s | M_t = m),
\end{aligned}
$$

for $s \in \{1, 2, ..., 2I\}, k \in \mathcal{M} = \{1, 2, ..., M\}$. It follows from Lemma 5, Lemma 8 and Theorem 3 in [8] that the spectral estimators $\hat{\mu}, \hat{P}$ have the following guarantee: pick any $\delta \in (0, 1)$, when the number of samples $n$ satisfies

$$
n \geq N_0 := \left( \frac{G^{\frac{2\sqrt{2}+1}{1-\theta}}}{\omega_{\min} \sigma^2} \right)^2 \log \left( \frac{2(S^2 + S)}{\delta} \right) \max \left\{ \frac{16 \times M^{1/3}}{C_0^{2/3} \omega_{\min}^{1/3}}, \frac{2\sqrt{2}M}{C_0^2 \omega_{\min} \sigma^2}, 1 \right\},
$$

then with probability $1 - \delta$ we have

$$
\| (\boldsymbol{\mu})^m - (\hat{\boldsymbol{\mu}})^m \|_2 \leq C_1 \sqrt{\frac{\log(6\frac{S^2+S}{\delta})}{n}},
$$

$$
\| P - \hat{P} \|_2 \leq C_2 \sqrt{\frac{\log(6\frac{S^2+S}{\delta})}{n}},
$$

for $m \in \mathcal{M}$ up to permutation, with

$$C_1 = \frac{21}{\sigma_{1,-1}} I C_3,$$

$$C_2 = \frac{4}{\sigma_{\min}(A_2)} \left( \sqrt{M} + M \frac{21}{\sigma_{1,-1}} \right) C_3,$$

$$C_3 = 2G \frac{2\sqrt{2}+1}{(1-\theta)\omega_{\min}^{0.5}} \left( 1 + \frac{8\sqrt{2}}{\omega_{\min}^2 \sigma^3} + \frac{256}{\omega_{\min}^2 \sigma^2} \right),$$

where $S = 2I$, $C_0$ is a numerical constant (see Theorem 16 in [8]), $\sigma_{1,-1}$ is the smallest nonzero singular value of the covariance matrix $\mathbb{E}[y_{t+1} \otimes y_{t-1}]$, and $\sigma = \min\{\sigma_{\min}(A_1), \sigma_{\min}(A_2), \sigma_{\min}(A_3)\}$, where $\sigma_{\min}(A_i)$ represents the smallest nonzero singular value of the matrix $A_i$, for $i = 1, 2, 3$. In addition, $\omega = (\omega(m))$ represents the stationary distribution of the underlying Markov chain $\{M_t\}$, and $\omega_{\min} = \min_m \omega(m) \geq \epsilon = \min_{i,j} P_{ij}$. Finally, $G$ and $\theta$ are the mixing rate parameters such that

$$\sup_{m_1} ||f_{1 \to t}(\cdot|m_1) - \omega||_{TV} \leq G\theta^{t-1},$$

where $f_{1 \to t}(\cdot|m_1)$ denotes the probability distribution vector of the underlying state at time $t$, starting from the initial state $m_1$. Under Assumption 3, one can take $G = 2$ and have the (crude) bound $\theta \leq 1 - \epsilon$, see e.g. Theorems 2.7.2 and 2.7.4 in [26].

## C   Proof of Proposition 2

*Proof.* We first introduce a few notations. Let $V_\beta(b)$ be the (optimal) value function of the infinite-horizon discounted version of the POMDP (or belief MDP) with discount factor $\beta \in (0,1)$ and initial belief state $b$. Define $v_\beta(b) := V_\beta(b) - V_\beta(s)$ for a fixed belief state $s$, where $v_\beta$ is the bias function for the *discounted* problem. We also introduce $\ell_1$ distance to the belief space $\mathcal{B}$: $\rho_b(b, b') := \|b - b'\|_1$. For any function $f : \mathcal{B} \mapsto \mathbb{R}$, define the Lipschitz module of a function $f$ by

$$l_{\rho_\mathcal{B}}(f) := \sup_{b \neq b'} \frac{|f(b) - f(b')|}{\rho_b(b, b')}.$$

The main idea of the proof is as follows. To bound $\text{span}(v) := \max_{b \in \mathcal{B}} v(b) - \min_{b \in \mathcal{B}} v(b)$ where $v$ is the bias function for our *undiscounted* problem, it suffices to bound the Lipschitz module of $v$ due to the fact that $\sup_{b \neq b'} ||b - b'||_1 = 2$. Under our assumptions, it can be shown that the bias function $v$ for the undiscounted problem satisfies the relation

$$v(b) = \lim_{\beta \to 1-} v_\beta(b), \quad \text{for } b \in \mathcal{B}. \tag{5}$$

Then applying Lemma 3.2(a) [23] yields

$$l_{\rho_\mathcal{B}}(v) \leq \lim_{\beta \to 1-} l_{\rho_\mathcal{B}}(v_\beta) = \lim_{\beta \to 1-} l_{\rho_\mathcal{B}}(V_\beta). \tag{6}$$

So it suffices to bound $\lim_{\beta \to 1-} l_{\rho_\mathcal{B}}(V_\beta)$. The bound for $l_{\rho_\mathcal{B}}(V_\beta)$ in turn implies that $(v_\beta)_\beta$ is a uniformly bounded equicontinuous family of functions, and hence validates (5) by Theorem 2 in [40].

We next proceed to bound $l_{\rho_\mathcal{B}}(V_\beta)$ and we will show that for any $\beta \in (0,1)$ we have $l_{\rho_\mathcal{B}}(V_\beta) \leq \frac{\eta}{1-\gamma}$ for some constants $\eta > 0, \gamma \in (0,1)$ that are both independent of $\beta$. To this end, we consider the finite horizon discounted belief MDP, and let $V_{n,\beta}$ be the optimal value function for the discounted problem with horizon length $n$ and discount factor $\beta$. Since the reward is bounded, it is readily seen that $\lim_{n \to \infty} V_{n,\beta} = V_\beta$. Then Lemma 2.1(e) [23] suggests that

$$l_{\rho_b}(V_\beta) \leq \liminf_{n \to \infty} l_{\rho_b}(V_{n,\beta}). \tag{7}$$

Thus, to bound $l_{\rho_b}(V_\beta)$, it suffices to bound the Lipschitz module $l_{\rho_b}(V_{n,\beta})$. The strategy is to apply the results including Lemmas 3.2 and 3.4 in [23], but it requires a new analysis to verify the conditions there.

To proceed, standard dynamic programming theory states that $V_{n,\beta}(b) = J_1(b)$, and $J_1(b)$ can be computed by the backward recursion:

$$J_n(b_n) = \bar{c}(b_n, i_n),$$

$$J_t(b_t) = \max_{i_t \in \mathcal{I}} \left\{ \bar{c}(b_t, i_t) + \beta \int_{\mathcal{B}} J_{t+1}(b_{t+1}) \bar{T}(db_{t+1}|b_t, i_t) \right\}, \quad 1 \le t < n, \tag{8}$$

where $\bar{T}$ is the (action-dependent) one-step transition law of the belief state, and $J_t(b_t)$ are finite for each $t$. More generally, for a given sequence of actions $i_{1:n}$, the $n$-step transition kernel for the belief state is define by

$$\bar{T}^{(n)}(\mathbf{A}|b, i_{1:n}) := \mathbb{P}(b_n \in \mathbf{A}|b_1 = b, i_{1:n}), \quad \mathbf{A} \subset \mathcal{B}. \tag{9}$$

To use the results in [23], we need to study the Lipschitz property of this multi-step transition kernel as we will see later. Following [23], we introduce the Lipschitz module for a transition kernel $\phi(b, db')$ on belief states. Let $K_{\rho_{\mathcal{B}}}(\nu, \theta)$ be the Kantorovich metric of two probability measure $\nu, \theta$ defined on $\mathcal{B}$:

$$K_{\rho_{\mathcal{B}}}(\nu, \theta) := \sup_f \left\{ \left| \int_{\mathcal{B}} f(b)\nu(db) - \int_{\mathcal{B}} f(b)\theta(db) \right|, f \in \mathrm{Lip}_1(\rho_{\mathcal{B}}) \right\},$$

where $\mathrm{Lip}_1(\rho_{\mathcal{B}})$ is the set of functions on $\mathcal{B}$ with Lipschitz module $l_{\rho_{\mathcal{B}}}(f) \le 1$. Then the Lipschitz module of the transition kernel $l_{\rho_{\mathcal{B}}}(\phi)$ is defined as:

$$l_{\rho_{\mathcal{B}}}(\phi) := \sup_{b^1 \ne b^2} \frac{K_{\rho_{\mathcal{B}}}(\phi(b^1, db'), \phi(b^2, db'))}{\rho_{\mathcal{B}}(b^1, b^2)}.$$

The transition kernel $\phi$ is called Lipschitz continuous if $l_{\rho_{\mathcal{B}}}(\phi) < \infty$. To bound $l_{\rho_{\mathcal{B}}}(V_{n,\beta})$ and to apply the results in [23], the key technical result we need is the following lemma. We defer its proof to the end of this section. Recall that $\epsilon = \min_{i,j \in \mathcal{M}} P_{i,j} > 0$.

**Lemma 1.** *For $1 \le n < \infty$, the $n$-step belief state transition kernel $\bar{T}^{(n)}(\cdot|b, i_{1:n})$ in (9) is uniformly Lipschitz in $i_{1:n}$, and the Lipschitz module is bounded as follows:*

$$l_{\rho_{\mathcal{B}}}(\bar{T}^{(n)}) \le C_4 \alpha^n + C_5,$$

*where $C_4 = \frac{2}{1-\alpha}$ and $C_5 = \frac{1}{2} + \frac{\alpha}{2}$ with $\alpha = 1 - \frac{\epsilon}{1-\epsilon} \in (0, 1)$. As a consequence, there exist constants $n_0 \in \mathbb{Z}^+$ and $\gamma < 1$ such that $l_{\rho_{\mathcal{B}}}(\bar{T}^{(n_0)}) < \gamma$ for any $i_{1:n}$. Here, we can take $n_0 = \lceil \log_\alpha \frac{1-C_5}{2C_4} \rceil$, and $\gamma = \frac{1}{2}(1 + C_5) = \frac{3+\alpha}{4}$.*

With Lemma 1, we are now ready to bound $l_{\rho_{\mathcal{B}}}(V_{n,\beta})$. Consider $n = kn_0$ for some positive integer $k$. We can infer from the value iteration in (8) that

$$J_t(b_t) = \sup_{i_{t:t+n_0-1}} \left\{ \sum_{l=0}^{n_0-1} \beta^l \int_{\mathcal{B}} \bar{c}(b_{t+l}, i_{t+l-1}) \bar{T}^{(l)}(db_{t+l}|b_t, i_{t:t+l-1}) \right.$$

$$\left. + \beta^{n_0} \int_{\mathcal{B}} J_{t+n_0}(b_{t+n_0}) \bar{T}^{(n_0)}(db_{t+n_0}|b_t, i_{t:t+n_0-1}) \right\}, \quad 1 \le t \le n - n_0 \tag{10}$$

Bounding $\bar{c}$ in (10) by $r_{\max} = 1$ (the bound for rewards) and $\bar{T}^{(l)}$ by its Lipschitz module, we obtain the following inequality using Lemmas 3.2 and 3.4 in [23]:

$$l_{\rho_{\mathcal{B}}}(J_t) \le r_{\max} \cdot \sum_{l=0}^{n_0-1} \beta^t l_{\rho_{\mathcal{B}}}^{\mathcal{I}^l}(\bar{T}^{(l)}) + \beta^{n_0} \cdot l_{\rho_{\mathcal{B}}}^{\mathcal{I}^{n_0}}(\bar{T}^{(n_0)}) \cdot l_{\rho_{\mathcal{B}}}(J_{t+n_0}),$$

where $l_{\rho_{\mathcal{B}}}^{\mathcal{I}^l}(\bar{T}^{(l)})$ is the supremum of the Lipschitz module $l_{\rho_{\mathcal{B}}}(\bar{T}^{(l)})$ over actions:

$$l_{\rho_{\mathcal{B}}}^{\mathcal{I}^l}(\bar{T}^{(l)}) := \sup_{i_{t:t+l-1}} \sup_{b_t \ne b_t'} \frac{K_{\rho_{\mathcal{B}}}(\bar{T}^{(l)}(db_{t+l}|b_t, i_{t:t+l-1}), \bar{T}^{(l)}(db_{t+l}|b_t', i_{t:t+l-1}))}{\rho_{\mathcal{B}}(b_t, b_t')}, \quad 0 \le l \le n_0.$$

Note that the value function in the last period $J_n(b_n) = \bar{c}(b_n, i_n)$ is uniformly Lipschitz in $i_n$ with Lipschitz module $r_{\max} = 1$. Applying the last inequality iteratively for $n_i = 1 + i n_0$ with $0 \le i < k$ and by Lemma 1, we have

$$
\begin{aligned}
l_{\rho\mathcal{B}}(J_{n_i}) &\le \sum_{t=0}^{n_0-1} \beta^t l_{\rho\mathcal{B}}^{\mathcal{I}^t}(\bar{T}^{(t)}) + \beta^{n_0} \cdot \gamma \cdot l_{\rho\mathcal{B}}(J_{n_{i+1}}) \\
&\le \sum_{t=0}^{n_0-1} [C_4 \alpha^t + C_5] + \beta^{n_0} \cdot \gamma \cdot l_{\rho\mathcal{B}}(J_{n_{i+1}}) \\
&\le \eta + \beta^{n_0} \gamma \cdot l_{\rho\mathcal{B}}(J_{n_{i+1}}),
\end{aligned}
$$

where

$$
\eta = \frac{C_4}{1-\alpha} + C_5 n_0, \tag{11}
$$

and $C_4, C_5, n_0, \alpha$ are given in Lemma 1. Iterating over $i$ and using $l_{\rho\mathcal{B}}(J_n) = l_{\rho\mathcal{B}}(J_{kn_0}) = r_{\max}$, we obtain

$$
l_{\rho\mathcal{B}}(J_0) \le \eta \cdot \frac{1 - (\beta^{n_0}\gamma)^k}{1 - \beta^{n_0}\gamma} + (\beta^{n_0}\gamma)^k \cdot r_{\max}.
$$

Recall that for $n = kn_0$, $V_{n,\beta}(b) = V_{kn_0,\beta}(b) = J_0(b)$. Since $\beta < 1$ and $\gamma < 1$, we then get

$$
\liminf_{k\to\infty} l_{\rho\mathcal{B}}(V_{kn_0,\beta}) \le \frac{\eta}{1-\gamma}.
$$

Together with (6) and (7), we can deduce that for a belief MDP satisfying $\min_{i,j\in\mathcal{M}} P_{ij} = \epsilon > 0$, the span of the bias function is upper bounded by

$$
span(v) \le D(\epsilon) := \frac{2\eta(\epsilon)}{1 - \gamma(\epsilon)},
$$

where with slight abuse of notations we use $\eta(\epsilon)$ (see (11)) and $\gamma(\epsilon)$ (see Lemma 1) to emphasize their dependency on $\epsilon$. The proof is completed by simplifying the expression of $D(\epsilon)$. $\quad\square$

## C.1 Proof of Lemma 1

*Proof.* Rewriting the Kantorovich metric, we have:

$$
\begin{aligned}
&K\{\bar{T}^{(n)}(db'|b^1, i_{1:n}), \bar{T}^{(n)}(db'|b^2, i_{1:n})\} \\
&= \sup_f \left\{ \left| \int f(b')\bar{T}^{(n)}(db'|b^1, i_{1:n}) - \int f(b')\bar{T}^{(n)}(db'|b^2, i_{1:n}) \right|, f \in \text{Lip}_1 \right\} \\
&= \sup_f \left\{ \left| \int f(b')\bar{T}^{(n)}(db'|b^1, i_{1:n}) - \int f(b')\bar{T}^{(n)}(db'|b^2, i_{1:n}) \right|, f \in \text{Lip}_1, ||f||_\infty \le 1 \right\}.
\end{aligned}
$$

The last equality follows from the following argument. Note that the span of a function $f$ with Lipschitz module 1 is bounded by $\text{Diam}(\mathcal{B})$ where $\text{Diam}(\mathcal{B}) := \sup_{b^1 \ne b^2} ||b^1 - b^2||_1 = 2$. So for any $f \in \text{Lip}_1$ we can find a constant $c$ that $||f + c||_\infty \le \text{Diam}(\mathcal{B})/2$. Moreover, let $\phi(f) = |\int f(b')\bar{T}^{(n)}(db'|b^1, i_{1:n}) - \int f(b')\bar{T}^{(n)}(db'|b^2, i_{1:n})|$, we know $\phi(f) = \phi(f + c)$ for any constant $c$. Without loss of generality, we can constrain $||f||_\infty \le \text{Diam}(\mathcal{B})/2 \le 1$.

We introduce a few notations to facilitate the presentation. We define the $n$-step reward kernel $\bar{Q}^{(n)}$, where $\bar{Q}^{(n)}(\prod_{t=1}^n dr_t|b, i_{1:n})$ is a probability measure on $\mathcal{R}^n$:

$$
\bar{Q}^{(n)}(A_1 \times ... \times A_n|b, i_{1:n}) = \mathbb{P}((r_1, \ldots, r_n) \in A_1 \times ... \times A_n|b, i_{1:n}).
$$

Given the initial belief $b$, we can define the $n$-step forward kernel $H^{(n)}$ as follows where $b_{n+1}$ is the belief at time $n + 1$:

$$
b_{n+1} = H^{(n)}(b, i_{1:n}, r_{1:n}).
$$

Then it is easy to see that the belief transition kernel $\bar{T}^{(n)}$ defined in (9) satisfies

$$\bar{T}^{(n)}(\mathbf{A}|b, i_{1:n}) = \int_{\mathcal{R}^n} \mathbb{1}_{\{H^{(n)}(b, i_{1:n}, r_{1:n}) \in \mathbf{A}\}} \bar{Q}^{(n)}(\prod_{t=1}^n dr_t | b, i_{1:n}).$$

Then we can obtain:

$$\left| \int_{\mathcal{R}^n} f(b') \bar{T}^{(n)}(db'|b^1, i_{1:n}) - \int_{\mathcal{R}^n} f(b') \bar{T}^{(n)}(db'|b^2, i_{1:n}) \right|$$

$$= \left| \int_{\mathcal{R}^n} f(H^{(n)}(b^1, i_{1:n}, r_{1:n})) \bar{Q}^{(n)}(\prod_{t=1}^n dr_t | b^1, i_{1:n}) - \int_{\mathcal{R}^n} f(H^{(n)}(b^2, i_{1:n}, r_{1:n})) \bar{Q}^{(n)}(\prod_{t=1}^n dr_t | b^2, i_{1:n}) \right|$$

$$\leq \left| \int_{\mathcal{R}^n} f(H^{(n)}(b^1, i_{1:n}, r_{1:n})) \left( \bar{Q}^{(n)}(\prod_{i=0}^{n-1} dr_t | b^1, i_{1:n}) - \bar{Q}^{(n)}(\prod_{t=1}^n dr_t | b^2, i_{1:n}) \right) \right|$$

$$+ \left| \int_{\mathcal{R}^n} \left( f(H^{(n)}(b^1, i_{1:n}, r_{1:n})) - f(H^{(n)}(b^2, i_{1:n}, r_{1:n})) \right) \bar{Q}^{(n)}(\prod_{t=1}^n dr_t | b^2, i_{1:n}) \right|. \tag{12}$$

We first bound the second term in (12). We can infer from Theorem 3.7.1 in [26] and its proof that the impact of initial belief decays exponentially fast:

$$|H^{(n)}(b^1, i_{1:n}, r_{1:n}) - H^{(n)}(b^2, i_{1:n}, r_{1:n})| \leq C_4 \alpha^n ||b^1 - b^2||_1,$$

where constant $C_4 = \frac{2(1-\epsilon)}{\epsilon}$ and $\alpha = \frac{1-2\epsilon}{1-\epsilon} < 1$. So the second term of (12) can be bounded by

$$\left| \int_{\mathcal{R}^n} \left( f(H^{(n)}(b^1, i_{1:n}, r_{1:n})) - f(H^{(n)}(b^2, i_{1:n}, r_{1:n})) \right) \bar{Q}^{(n)}(\prod_{t=1}^n dr_t | b^2, i_{1:n}) \right|$$

$$\leq \left| \int_{\mathcal{R}^n} \left| H^{(n)}(b^1, i_{1:n}, r_{1:n}) - H^{(n)}(b^2, i_{1:n}, r_{1:n}) \right| \bar{Q}^{(n)}(\prod_{t=1}^n dr_t | b^2, i_{1:n}) \right|$$

$$\leq C_4 \alpha^n ||b^1 - b^2||_1, \tag{13}$$

where the first inequality follows from $f \in \text{Lip}_1$.

It remains to bound the first term in (12). Recall that $b$ defines the initial probability distribution $M_1$. The $n$ steps observation kernel is

$$\bar{Q}^{(n)}(\prod_{t=1}^n dr_t | b, i_{1:n}) = \sum_{m \in \mathcal{M}} \mathbb{P}(M_1 = m) \mathbb{P}(\prod_{t=1}^n dr_t | M_1 = m, i_{1:n}) = \sum_{m \in \mathcal{M}} b(m) \mathbb{P}(\prod_{t=1}^n dr_t | M_1 = m, i_{1:n}).$$

Define a vector $g \in \mathbb{R}^M$ as:

$$g(m) := \int_{\mathcal{R}^n} f(H^{(n)}(b^1, i_{1:n}, r_{1:n})) \mathbb{P}(\prod_{t=1}^n dr_t | M_1 = m, i_{1:n}).$$

We can rewrite the first term of (12):

$$\left| \int_{\mathcal{R}^n} f(H^{(n)}(b^1, i_{1:n}, r_{1:n})) \left( \bar{Q}^{(n)}(\prod_{t=1}^n dr_t | b^1, i_{1:n}) - \bar{Q}^{(n)}(\prod_{t=1}^n dr_t | b^2, i_{1:n}) \right) \right|$$

$$= \left| \sum_{m=1}^M (b^1(m) - b^2(m)) \int_{\mathcal{R}^n} f(H^{(n)}(b^1, i_{1:n}, r_{1:n})) \mathbb{P}(\prod_{t=1}^n dr_t | M_1 = m, i_{1:n}) \right|$$

$$= \left| \sum_{m=1}^M \left( b^1(m) - b^2(m) \right) g(m) \right|$$

$$= \left| \sum_{m=1}^M \left( b^1(m) - b^2(m) \right) \left( g(m) - \frac{\max_m g(m) + \min_m g(m)}{2} \right) \right|$$

$$\leq \left\| b^1 - b^2 \right\|_1 \cdot \left\| g(m) - \frac{\max_m g(m) + \min_m g(m)}{2} \right\|_\infty$$

$$= \left\| b^1 - b^2 \right\|_1 \frac{1}{2} \left( \max_m g(m) - \min_m g(m) \right), \tag{14}$$

where the next to last equality follows from $\sum_{m=1}^M \left( b^1(m) - b^2(m) \right) = 0$.

Next we bound $\max_m g(m) - \min_m g(m)$. From the equation above, it is clear that the quantity $\frac{1}{2} \left( \max_m g(m) - \min_m g(m) \right) \leq 1$, because $||f||_\infty \leq 1$. However to prove Lemma 1, we need a sharper bound so that we can find a constant $C_5 < 1$ (that is independent of $b^1$, $n$ and $i_{1:n}$) with

$$\frac{1}{2} \left( \max_m g(m) - \min_m g(m) \right) \leq C_5 < 1. \tag{15}$$

Suppose (15) holds. Then on combining (12), (13) and (14), we obtain

$$\left| \int_{\mathcal{B}} f(b') \bar{T}^{(n)}(db' | b^1, i_{1:n}) - \int_{\mathcal{B}} f(b') \bar{T}^{(n)}(db' | b^2, i_{1:n}) \right| \leq C_4 \alpha^n ||b^1 - b^2||_1 + C_5 ||b^1 - b^2||_1.$$

It then follows that the Kantorovich metric is bounded by

$$K \left( \bar{T}^{(n)}(db' | b^1, i_{1:n}), \bar{T}^{(n)}(db' | b^2, i_{1:n}) \right) \leq C_4 \alpha^n ||b^1 - b^2||_1 + C_5 ||b^1 - b^2||_1,$$

where $C_4 = \frac{2(1-\epsilon)}{\epsilon}, \alpha = 1 - \frac{\epsilon}{1-\epsilon}$, and $\epsilon = \min_{m, m' \in \mathcal{M}} P_{m, m'} > 0$. So $\bar{T}^{(n)}$ is Lipschitz uniformly in actions, and its Lipschitz module can be bounded as follows:

$$l_{\rho_{\mathcal{B}}}^{\mathcal{I}^n}(\bar{T}^{(n)}) := \sup_{i_{1:n}} \sup_{b^1 \neq b^2} \frac{K \left( \bar{T}^{(n)}(db' | b^1, i_{1:n}), \bar{T}^{(n)}(db' | b^2, i_{1:n}) \right)}{\rho_{\mathcal{B}}(b^1, b^2)} \leq C_4 \alpha^n + C_5.$$

If we choose $n = n_0 := \lceil \log_\alpha \frac{1 - C_5}{2 C_4} \rceil$, so that $C_4 \alpha^{n_0} + C_5 < \frac{1}{2}(1 + C_5) := \gamma < 1$, then we obtain the desired result $l_{\rho_{\mathcal{B}}}^{\mathcal{I}^{n_0}}(\bar{T}^{(n_0)}) < \gamma$.

It remains to prove (15). Since the set $\mathcal{M} = \{1, \ldots, M\}$ is finite, we pick $m^* \in \underset{m \in \mathcal{M}}{\operatorname{argmin}} \, g(m), \hat{m} \in \underset{m \in \mathcal{M}}{\operatorname{argmax}} \, g(m)$. We have

$$\frac{1}{2} \left( \max_m g(m) - \min_m g(m) \right) \tag{16}$$

$$= \frac{1}{2} \sum_{r_{1:n} \in \mathcal{R}^n} f(H^{(n)}(b^1, i_{1:n}, r_{1:n})) \left( \mathbb{P}(r_{1:n} | M_1 = \hat{m}, i_{1:n}) - \mathbb{P}(r_{1:n} | M_1 = m^*, i_{1:n}) \right)$$

$$\leq \frac{1}{2} \sum_{r_{1:n} \in \mathcal{R}^n} \left| \mathbb{P}(r_{1:n} | M_1 = \hat{m}, i_{1:n}) - \mathbb{P}(r_{1:n} | M_1 = m^*, i_{1:n}) \right|,$$

where the inequality follows from Hölder's inequality with $||f||_\infty \le 1$. We can compute

$$\mathbb{P}(r_{1:n}|M_1 = m_1, i_{1:n})$$

$$= \sum_{m_{2:n} \in \mathcal{M}^{n-1}} \mathbb{P}(r_{1:n}|M_1 = m_1, i_{1:n}, M_{2:n} = m_{2:n}) \cdot \mathbb{P}(M_{2:n} = m_{2:n}|M_1 = m_1, i_{1:n})$$

$$= \sum_{m_{2:n} \in \mathcal{M}^{n-1}} \mathbb{P}(r_{1:n}|i_{1:n}, m_{1:n}) \cdot \mathbb{P}(m_{2:n}|M_1 = m_1, i_{1:n})$$

$$= \sum_{m_{2:n} \in \mathcal{M}^{n-1}} \left( \prod_{t=1}^{n} \mathbb{P}(r_t|m_t, i_t) \right) \cdot \left( \prod_{t=1}^{n-1} \mathbb{P}(m_{t+1}|m_t, i_t) \right),$$

where the last equality holds due to the conditional independence. We can then infer that for any $\{r_{1:n}\}, \{i_{1:n}\}$,

$$\mathbb{P}(r_{1:n}|M_1 = m^*, i_{1:n})$$

$$= \sum_{m_{2:n} \in \mathcal{M}^{n-1}} \left( \prod_{t=2}^{n} \mathbb{P}(r_t|m_t, i_t) \right) \cdot \left( \prod_{t=1}^{n-1} \mathbb{P}(m_{t+1}|m_t, i_t) \right) \cdot P(m^*, m_2)$$

$$\ge \sum_{m_{2:n} \in \mathcal{M}^{n-1}} \left( \prod_{t=1}^{n} \mathbb{P}(r_t|m_t, i_t) \right) \cdot \left( \prod_{t=2}^{n-1} \mathbb{P}(m_{t+1}|m_t, i_t) \right) \cdot P(\hat{m}, m_1) \cdot \frac{\epsilon}{1-\epsilon}$$

$$= \mathbb{P}(r_{1:n}|M_1 = \hat{m}, i_{1:n}) \cdot \frac{\epsilon}{1-\epsilon}.$$

It follows that

$$|\mathbb{P}(r_{1:n}|M_1 = \hat{m}, i_{1:n}) - \mathbb{P}(r_{1:n}|M_1 = m^*, i_{1:n})|$$

$$\le \max\left\{ \left(1 - \frac{\epsilon}{1-\epsilon}\right) \mathbb{P}(r_{1:n}|M_1 = \hat{m}, i_{1:n}), \mathbb{P}(r_{1:n}|M_1 = m^*, i_{1:n}) \right\}$$

$$\le \left(1 - \frac{\epsilon}{1-\epsilon}\right) \mathbb{P}(r_{1:n}|M_1 = \hat{m}, i_{1:n}) + \mathbb{P}(r_{1:n}|M_1 = m^*, i_{1:n}).$$

Then we can obtain from (16) that

$$\frac{1}{2}\left( \max_m g(m) - \min_m g(m) \right)$$

$$\le \frac{1}{2} \sum_{r_{1:n} \in \mathcal{R}^n} \left[ \left(1 - \frac{\epsilon}{1-\epsilon}\right) \mathbb{P}(r_{1:n}|M_0 = \hat{m}, i_{1:n}) + \mathbb{P}(r_{1:n}|M_0 = m^*, i_{1:n}) \right]$$

$$= \alpha/2 + 1/2 := C_5 < 1,$$

where $\alpha = 1 - \frac{\epsilon}{1-\epsilon} \in (0, 1)$. The proof is complete. $\qquad\square$

## D  Proof of Proposition 3

The proof of Proposition 3 largely follows the proof of Proposition 3 in [18], with minor changes to take into account the action sequence, so we omit details.

## E  Proof of Theorem 1

In this section we provide a complete proof of Theorem 1. We follow our prior simplication that random reward follows the Bernoulli distribution. Here we want to clarify two notations in advance. $\mathbb{E}^\pi$ means the expectation is taken respect to true mean reward matrix $\boldsymbol{\mu}$ and transition probabilities $P$, and $\mathbb{E}_k^\pi$ denotes that the underlying parameters are estimators $\boldsymbol{\mu}_k$ and $P_k$.

Recalling the definition of regret in (1), it can be rewritten as:

$$\mathcal{R}_T = \sum_{t=1}^{T} (\rho^* - R_t) = \sum_{t=1}^{T} (\rho^* - \mathbb{E}^\pi[R_t|\mathcal{F}_{t-1}]) + \sum_{t=1}^{T} (\mathbb{E}^\pi[R_t|\mathcal{F}_{t-1}] - R_t). \qquad (17)$$

We first bound the second term of (17), i.e. the total bias between the conditional expectation of reward and the realization. Define a stochastic process $\{X_n, n = 0, \cdots, T\}$ as:

$$X_0 = 0, \quad X_t = \sum_{l=1}^{t} (\mathbb{E}^\pi[R_l|\mathcal{F}_{l-1}] - R_l),$$

then the second term in (17) is $X_T$. It is easy to see that $X_t$ is a martingale. Moreover, due to the Bernoulli distribution of $R_t$,

$$|X_{t+1} - X_t| = |\mathbb{E}^\pi[R_{t+1}|\mathcal{F}_t] - R_{t+1}| \le 1.$$

Applying the Azuma-Hoeffding inequality [9], we have

$$\mathbb{P}\left(\sum_{t=1}^{T}(\mathbb{E}^\pi[R_t|\mathcal{F}_{t-1}] - R_t) \ge \sqrt{2T \ln \frac{1}{\delta}}\right) \le \delta. \tag{18}$$

Next we bound the first term of (17). Recall that the definition of belief state under the optimistic and true parameters: $b_t^k(m) = \mathbb{P}_{\boldsymbol{\mu}_k, P_k}(M_t = m|\mathcal{F}_{t-1})$ and $b_t(m) = \mathbb{P}(M_t = m|\mathcal{F}_{t-1})$, and the definition of reward functions with respect to the true belief state $\bar{c}(b_t, i) = \sum_{m=1}^{M} \mu_{m,i} b_t(m)$. We can also define the reward functions with respect to the optimistic belief state $b_t^k$ as:

$$\bar{c}_k(b_t^k, i) = \sum_{m=1}^{M} (\boldsymbol{\mu}_k)_{m,i} b_t^k(m).$$

Because $I_t$ is also adapted to $\mathcal{F}_{t-1}$, we have

$$\mathbb{E}^\pi[\mu(M_t, I_t)|\mathcal{F}_{t-1}] = \bar{c}(b_t, I_t) = \langle (\boldsymbol{\mu})_{I_t}, b_t \rangle,$$
$$\mathbb{E}_k^\pi[\mu_k(M_t, I_t)|\mathcal{F}_{t-1}] = \bar{c}_k(b_t^k, I_t) = \langle (\boldsymbol{\mu}_k)_{I_t}, b_t^k \rangle, \tag{19}$$

where $\mu(M_t, I_t), \mu_k(M_t, I_t)$ are the $M_t$-th row $I_t$-th column element of matrix $\mu, \mu_k$ respectively, and $(\boldsymbol{\mu}_k)_{I_t}$ and $(\boldsymbol{\mu})_{I_t}$ are the $I_t$-th column vector of the reward matrix $\boldsymbol{\mu}_k$ and $\boldsymbol{\mu}$, respectively.

Then we can rewrite the first term of (17):

$$\sum_{t=1}^{T}(\rho^* - \mathbb{E}^\pi[R_t|\mathcal{F}_{t-1}]) = \sum_{t=1}^{T}(\rho^* - \mathbb{E}^\pi[\mu(M_t, I_t)|\mathcal{F}_{t-1}]) = \sum_{t=1}^{T}(\rho^* - \bar{c}(b_t, I_t)), \tag{20}$$

where the first equation is due to the tower property and the fact that $R_t$ and $\mathcal{F}_{t-1}$ are conditionally independent given $M_t$ and $I_t$.

Let $K$ be the number of total episodes. For each episode $k = 1, 2, \cdots, K$, let $H_k, E_k$ be the exploration and exploitation phases, respectively. Then we can split equation (20) to the summation of the bias in these two phases as:

$$\sum_{k=1}^{K} \sum_{t \in H_k} (\rho^* - \bar{c}(b_t, I_t)) + \sum_{k=1}^{K} \sum_{t \in E_k} (\rho^* - \bar{c}(b_t, I_t)). \tag{21}$$

Moreover, we remark here that the length of the last exploitation phase $|E_K| = \min\{\tau_2\sqrt{K}, \max\{T - (K\tau_1 + \sum_{k=1}^{K-1} \tau_2\sqrt{k}), 0\}\}$, as it may end at period $T$.

*Step 1: Bounding the regret in exploration phases*

The first term of (21) can be simply upper bounded by:

$$\sum_{k=1}^{K} \sum_{t \in H_k} (\rho^* - \bar{c}(b_t, I_t)) \le \sum_{k=1}^{K} \sum_{t \in H_k} \rho^* = K\tau_1\rho^*. \tag{22}$$

*Step 2: Bounding the regret in exploitation phases*

We bound it by separating into "success" and "failure" events below. Recall that in episode $k$, we define the set of plausible POMDPs $\mathbb{G}_k(\delta_k)$, which is defined in terms of confidence regions $\mathcal{C}_k(\delta_k)$

around the estimated mean reward matrix $\boldsymbol{\mu}_k$ and the transition probabilities $P_k$. Then choose an optimistic POMDP $\widetilde{\mathbb{G}}_k \in \mathbb{G}_k(\delta_k)$ that has the optimal average reward among the plausible POMDPs and denote its corresponding reward matrix, value function, and the optimal average reward by $\boldsymbol{\mu}_k, v_k$ and $\rho^k$, respectively. Thus, we say a "success" event if and only if the set of plausible POMDPs $\mathbb{G}_k(\delta_k)$ contains the true POMDP $\mathbb{G}$. In the following proof, we omit the dependence on $\delta_k$ from $\mathbb{G}_k(\delta_k)$ for simplicity. From Algorithm 2, the confidence level of $\mu_k$ in episode $k$ is $1 - \delta_k$, we can obtain:

$$\mathbb{P}(\mathbb{G} \notin \mathbb{G}_k, \text{for some } k) \leq \sum_{k=1}^{K} \delta_k = \sum_{k=1}^{K} \frac{\delta}{k^3} \leq \frac{3}{2}\delta.$$

Thus, with probability at least $1 - \frac{3}{2}\delta$, "success" events happen. It means $\rho^* \leq \rho^k$ for any $k$ because $\rho^k$ is the optimal average reward of the optimistic POMDP $\widetilde{\mathbb{G}}_k$ from the set $\mathbb{G}_k$. Then we can bound the regret of "success" events in exploitation phases as follows.

$$\sum_{k=1}^{K} \sum_{t \in E_k} (\rho^* - \bar{c}(b_t, I_t)) \leq \sum_{k=1}^{K} \sum_{t \in E_k} (\rho^k - \bar{c}(b_t, I_t))$$

$$= \sum_{k=1}^{K} \sum_{t \in E_k} (\rho^k - \bar{c}_k(b_t^k, I_t)) + (\bar{c}_k(b_t^k, I_t) - \bar{c}(b_t, I_t)). \qquad (23)$$

To bound the first term of formula (23), we use the Bellman optimality equation for the optimistic belief MDP $\widetilde{\mathbb{G}}_k$ on the continuous belief state space $\mathcal{B}$:

$$\rho^k + v_k(b_t^k) = \bar{c}_k(b_t^k, I_t) + \int_{b_{t+1}^k \in \mathcal{B}} v_k(b_{t+1}^k) \bar{T}_k(db_{t+1}^k | b_t^k, I_t) = \bar{c}_k(b_t^k, I_t) + \langle \bar{T}_k(\cdot|b_t^k, I_t), v_k(\cdot) \rangle,$$

where $\bar{T}_k(\cdot|b_t^k, I_t) = \mathbb{P}_{\boldsymbol{\mu}_k, P_k}(b_{t+1} \in \cdot|b_t^k, I_t)$ means transition probability of the belief state conditional on pulled arm under estimated reward matrix $\boldsymbol{\mu}_k$ and transition matrix of underlying Markov chain $P_k$ at time $t$.

Moreover, we note that if value function $v_k$ satisfies the Bellman equation (2), then so is $v_k + c\mathbf{1}$. Thus, without loss of generality, we assume that $v_k$ needs to satisfy $||v_k||_\infty \leq \text{span}(v_k)/2$. Then from Proposition 2, suppose the uniform bound of $\text{span}(v_k)$ is $D$, then we have:

$$||v_k||_\infty \leq \frac{1}{2}\text{span}(v_k) \leq \frac{D}{2}. \qquad (24)$$

Thus the first term of (23) can be bounded by

$$\sum_{k=1}^{K} \sum_{t \in E_k} (\rho^k - \bar{c}_k(b_t^k, I_t))$$

$$= \sum_{k=1}^{K} \sum_{t \in E_k} (-v_k(b_t^k) + \langle \bar{T}_k(\cdot|b_t^k, I_t), v_k(\cdot) \rangle)$$

$$= \sum_{k=1}^{K} \sum_{t \in E_k} (-v_k(b_t^k) + \langle \bar{T}(\cdot|b_t^k, I_t), v_k(\cdot) \rangle) + \langle \bar{T}_k(\cdot|b_t^k, I_t) - \bar{T}(\cdot|b_t^k, I_t), v_k(\cdot) \rangle, \qquad (25)$$

where recall that $\bar{T}_k(\cdot|b_t^k, I_t)$ and $\bar{T}(\cdot|b_t^k, I_t)$ in the second equality are the belief state transition probabilities under estimated and true parameters, respectively. And the last inequality is applying the Hölder's inequality.

For the first term of (25), we have

$$\sum_{k=1}^{K} \sum_{t \in E_k} \left( -v_k(b_t^k) + \langle \bar{T}(\cdot | b_t^k, I_t), v_k(\cdot) \rangle \right)$$

$$= \sum_{k=1}^{K} \sum_{t \in E_k} \left( -v_k(b_t^k) + v_k(b_{t+1}^k) \right) + \left( -v_k(b_{t+1}^k) + \langle \bar{T}(\cdot | b_t^k, I_t), v_k(\cdot) \rangle \right)$$

$$= \sum_{k=1}^{K} v_k(b_{t_k+\tau_1+\tau_2\sqrt{k}}^k) - v_k(b_{t_k+\tau_1+1}^k) + \sum_{k=1}^{K} \sum_{t \in E_k} \mathbb{E}^{\pi}[v_k(b_{t+1}^k)|\mathcal{F}_t] - v_k(b_{t+1}^k).$$

where the first term in the last equality is due to the telescoping from $t_k + \tau_1 + 1$ to $t_k + \tau_1 + \tau_2\sqrt{k}$, the start and end of the exploitation phase in episode $k$. The second term in the last equality is because:

$$\langle \bar{T}(\cdot | b_t^k, I_t), v_k(\cdot) \rangle = \int_{b_{t+1}^k \in \mathcal{B}} v_k(b_{t+1}^k) \bar{T}(db_{t+1}^k | b_t^k, I_t) = \mathbb{E}^{\pi}[v_k(b_{t+1}^k)|b_t^k] = \mathbb{E}^{\pi}[v_k(b_{t+1}^k)|\mathcal{F}_t].$$

Applying Proposition 2, we have

$$v_k(b_{t_k+\tau_1+\tau_2\sqrt{k}}^k) - v_k(b_{t_k+\tau_1+1}^k) \le D.$$

We also need the following result, the proof of which is deferred to the end of this section.

**Proposition 4.** *Let $K$ be the number of total episodes up to time $T$. For each episode $k = 1, \cdots, K$, let $E_k$ be the index set of the $k$th exploitation phase and $v_k$ be the value function of the optimistic POMDP at the $k$th exploitation phase. Then with probability at most $\delta$,*

$$\sum_{k=1}^{K} \sum_{t \in E_k} \mathbb{E}^{\pi}[v_k(b_{t+1})|\mathcal{F}_t] - v_k(b_{t+1}) \ge D\sqrt{2T\ln(\frac{1}{\delta})},$$

*where the expectation $\mathbb{E}^{\pi}$ is taken respect to the true parameters $\boldsymbol{\mu}$ and $P$ under policy $\pi$, and the filtration $\mathcal{F}_t$ is defined as $\mathcal{F}_t := \sigma(\pi_1, R_1^{\pi}, ..., \pi_{t-1}, R_{t-1}^{\pi})$.*

Applying Proposition 4, with probability at least $1 - \delta$, we have:

$$\sum_{k=1}^{K} \sum_{t \in E_k} \mathbb{E}^{\pi}[v_k(b_{t+1}^k)|\mathcal{F}_t] - v_k(b_{t+1}^k) \le D\sqrt{2T\ln(\frac{1}{\delta})}.$$

Thus, the first term of (25) can be upper bounded by:

$$KD + D\sqrt{2T\ln(\frac{1}{\delta})}. \tag{26}$$

For the second term of (25), we note that $\bar{T}(b_{t+1}^k | b_t^k, I_t)$ is zero except for two points where $b_{t+1}^k$ are exactly the Bayesian updating after receiving an observation of Bernoulli reward $r_t$ taking value 0 or 1. Thus, we have the following transition kernel:

$$\langle \bar{T}_k(\cdot | b_t^k, I_t) - \bar{T}(\cdot | b_t^k, I_t), v_k(\cdot) \rangle$$

$$\le \left| \int_{\mathcal{B}} v_k(b') \bar{T}_k(db' | b_t^k, I_t) - \int_{\mathcal{B}} v_k(b') \bar{T}(db' | b_t^k, I_t) \right|$$

$$= \left| \sum_{r_t \in \mathcal{R}} v_k \left( H_k \left( b_t^k, I_t, r_t \right) \right) \mathbb{P}_k \left( r_t | b_t^k, I_t \right) - \sum_{r_t \in \mathcal{R}} v_k \left( H \left( b_t^k, I_t, r_t \right) \right) \mathbb{P} \left( r_t | b_t^k, I_t \right) \right|$$

$$\le \left| \sum_{r_t \in \mathcal{R}} v_k \left( H_k \left( b_t^k, I_t, r_t \right) \right) \cdot \left[ \mathbb{P}_k \left( r_t | b_t^k, I_t \right) - \mathbb{P} \left( r_t | b_t^k, I_t \right) \right] \right|$$

$$+ \left| \sum_{r_t \in \mathcal{R}} \left[ v_k \left( H_k \left( b_t^k, I_t, r_t \right) \right) - v_k \left( H \left( b_t^k, I_t, r_t \right) \right) \right] \cdot \mathbb{P} \left( r_t | b_t^k, I_t \right) \right|, \tag{27}$$

where we use $H_k$ and $H$ to denote the belief updating function under the optimistic model $(\boldsymbol{\mu}_k, P_k)$ and the true model $(\boldsymbol{\mu}, P)$, and we use $\mathbb{P}_k$ and $\mathbb{P}$ to denote the probability with respect to the optimistic model and true model respectively.

We bound the first term of (27) by

$$
\begin{aligned}
&\left| \sum_{r_t \in \mathcal{R}} v_k \left( H_k \left( b_t^k, I_t, r_t \right) \right) \cdot \left[ \mathbb{P}_k \left( r_t | b_t^k, I_t \right) - \mathbb{P} \left( r_t | b_t^k, I_t \right) \right] \right| \\
&\leq \left| v_k \left( H_k \left( b_t^k, I_t, r_t = 1 \right) \right) \cdot \left[ \mathbb{P}_k \left( r_t = 1 | b_t^k, I_t \right) - \mathbb{P} \left( r_t = 1 | b_t^k, I_t \right) \right] \right| \\
&\quad + \left| v_k \left( H_k \left( b_t^k, I_t, r_t = 0 \right) \right) \cdot \left[ \mathbb{P}_k \left( r_t = 0 | b_t^k, I_t \right) - \mathbb{P} \left( r_t = 0 | b_t^k, I_t \right) \right] \right| \\
&= \left| v_k \left( H_k \left( b_t^k, I_t, 1 \right) \right) \cdot \left[ \langle (\boldsymbol{\mu}_k)_{I_t}, b_t^k \rangle - \langle (\boldsymbol{\mu})_{I_t}, b_t^k \rangle \right] \right| \\
&\quad + \left| v_k \left( H_k \left( b_t^k, I_t, 0 \right) \right) \cdot \left[ 1 - \langle (\boldsymbol{\mu}_k)_{I_t}, b_t^k \rangle - \left( 1 - \langle (\boldsymbol{\mu})_{I_t}, b_t^k \rangle \right) \right] \right| \\
&\leq 2 \|v_k\|_\infty \cdot \left| \langle (\boldsymbol{\mu}_k)_{I_t}, b_t^k \rangle - \langle (\boldsymbol{\mu})_{I_t}, b_t^k \rangle \right| \\
&\leq D \left| \langle (\boldsymbol{\mu}_k)_{I_t}, b_t^k \rangle - \langle (\boldsymbol{\mu})_{I_t}, b_t^k \rangle \right| \\
&\leq D \| (\boldsymbol{\mu}_k)_{I_t} - (\boldsymbol{\mu})_{I_t} \|_1 \cdot \| b_t^k \|_\infty \\
&\leq D \| (\boldsymbol{\mu}_k)_{I_t} - (\boldsymbol{\mu})_{I_t} \|_1,
\end{aligned}
\tag{28}
$$

where the first equality comes from $\mathbb{P}_k \left( r_t = 1 | b_t^k, I_t \right) = \sum_{m \in \mathcal{M}} \mathbb{P}_k (r_t = 1 | m, I_t) b_t^k(m) = \langle (\boldsymbol{\mu})_{I_t}, b_t^k \rangle$ and $\mathbb{P}_k \left( r_t = 0 | b_t^k, I_t \right) = \sum_{m \in \mathcal{M}} \mathbb{P}_k (r_t = 0 | m, I_t) b_t^k(m) = 1 - \langle (\boldsymbol{\mu})_{I_t}, b_t^k \rangle$, and the third inequality is from (24), we have $\|v_k\|_\infty \leq \frac{D}{2}$.

We can bound the second term of (27) as follows:

$$
\begin{aligned}
&\left| \sum_{r_t \in \mathcal{R}} \left[ v_k \left( H_k \left( b_t^k, I_t, r_t \right) \right) - v_k \left( H \left( b_t^k, I_t, r_t \right) \right) \right] \cdot \mathbb{P} \left( r_t | b_t^k, I_t \right) \right| \\
&\leq \sum_{r_t \in \mathcal{R}} \left| v_k \left( H_k \left( b_t^k, I_t, r_t \right) \right) - v_k \left( H \left( b_t^k, I_t, r_t \right) \right) \right| \cdot \mathbb{P} \left( r_t | b_t^k, I_t \right) \\
&\leq \sum_{r_t \in \mathcal{R}} \frac{D}{2} \left| H_k \left( b_t^k, I_t, r_t \right) - H \left( b_t^k, I_t, r_t \right) \right| \cdot \mathbb{P} \left( r_t | b_t^k, I_t \right) \\
&\leq \sum_{r_t \in \mathcal{R}} \frac{D}{2} \left( L_1 \| \boldsymbol{\mu} - \boldsymbol{\mu}_k \|_1 + L_2 \| P - P_k \|_F \right) \cdot (1 - \alpha) \cdot \mathbb{P} \left( r_t | b_t^k, I_t \right) \\
&= \frac{D}{2} \left( L_1 \| \boldsymbol{\mu} - \boldsymbol{\mu}_k \|_1 + L_2 \| P - P_k \|_F \right) \cdot (1 - \alpha),
\end{aligned}
\tag{29}
$$

where $\alpha = 1 - \frac{\epsilon}{1 - \epsilon}$. Here, the second inequality is implied from the proof of Proposition 2. The last inequality follows from a refined bound in (4) of Proposition 3. Specifically, the current upper bound in Proposition 3 is uniform in time, and it is crude for small $t$ since the initial belief is the same with $b_1 = \hat{b}_1$. One can improve the bound by multiplying it with a factor of $1 - \alpha^{t-1}$. This follows from Proposition 3 of [18] and its proof. Therefore, we can obtain the last inequality of (29), since here we consider the one-step update of the beliefs under two different sets of model parameters.

Therefore, from (28) and (29), we can obtain that the second term of equation (25):

$$
\begin{aligned}
&\sum_{k=1}^K \sum_{t \in E_k} \langle \bar{T}_k(\cdot | b_t^k, I_t) - \bar{T}(\cdot | b_t^k, I_t), v_k(\cdot) \rangle \\
&\leq \sum_{k=1}^K \sum_{t \in E_k} D \left[ \| (\boldsymbol{\mu})_{I_t} - (\boldsymbol{\mu}_k)_{I_t} \|_1 + \frac{L_1(1 - \alpha)}{2} \| \boldsymbol{\mu} - \boldsymbol{\mu}_k \|_1 + \frac{L_2(1 - \alpha)}{2} \| P - P_k \|_F \right].
\end{aligned}
\tag{30}
$$

Summing up (26) and (30), the first term of formula (23) can be bounded by

$$\sum_{k=1}^{K}\sum_{t\in E_k}(\rho^k - \bar{c}_k(b_t^k, I_t)) \leq KD + D\sqrt{2T\ln(\frac{1}{\delta})}$$

$$+\sum_{k=1}^{K}\sum_{t\in E_k} D\left[||(\boldsymbol{\mu})_{I_t} - (\boldsymbol{\mu}_k)_{I_t}||_1 + \frac{L_1(1-\alpha)}{2}||\boldsymbol{\mu} - \boldsymbol{\mu}_k||_1 + \frac{L_2(1-\alpha)}{2}||P - P_k||_F\right]. (31)$$

Next we proceed to bound the second term of (23). By (19), it can be rewritten as

$$\sum_{k=1}^{K}\sum_{t\in E_k}\bar{c}_k(b_t^k, I_t) - \bar{c}(b_t, I_t)$$

$$=\sum_{k=1}^{K}\sum_{t\in E_k}\langle(\boldsymbol{\mu}_k)_{I_t}, b_t^k\rangle - \langle(\boldsymbol{\mu})_{I_t}, b_t\rangle$$

$$=\sum_{k=1}^{K}\sum_{t\in E_k}\langle(\boldsymbol{\mu}_k)_{I_t}, b_t^k\rangle - \langle(\boldsymbol{\mu})_{I_t}, b_t^k\rangle + \langle(\boldsymbol{\mu})_{I_t}, b_t^k\rangle - \langle(\boldsymbol{\mu})_{I_t}, b_t\rangle,$$

then from the Hölder's inequality, we can further bounded the right hand side of above formula to

$$\sum_{k=1}^{K}\sum_{t\in E_k}||(\boldsymbol{\mu}_k)_{I_t} - (\boldsymbol{\mu})_{I_t}||_1||b_t^k||_\infty + ||(\boldsymbol{\mu})_{I_t}||_\infty||b_t^k - b_t||_1$$

$$\leq \sum_{k=1}^{K}\sum_{t\in E_k}||(\boldsymbol{\mu}_k)_{I_t} - (\boldsymbol{\mu})_{I_t}||_1 + ||b_t^k - b_t||_1.$$

By Proposition 3, we have $||b_t^k - b_t||_1 \leq L_1||\boldsymbol{\mu} - \boldsymbol{\mu}_k||_1 + L_2||P - P_k||_F$. Combining the above expression with (31), we can see that with probability $1-\delta$, the regret incurred from "success" events, i.e., (23), can be bounded

$$\sum_{k=1}^{K}\sum_{t\in E_k}(\rho^* - \bar{c}(b_t, I_t))$$

$$\leq KD + D\sqrt{2T\ln(\frac{1}{\delta})} + \sum_{k=1}^{K}\sum_{t\in E_k}(D+1)||(\boldsymbol{\mu})_{I_t} - (\boldsymbol{\mu}_k)_{I_t}||_1$$

$$+\sum_{k=1}^{K}\sum_{t\in E_k}\left[\left(1 + \frac{D(1-\alpha)}{2}\right)L_1||\boldsymbol{\mu} - \boldsymbol{\mu}_k||_1 + \left(1 + \frac{D(1-\alpha)}{2}\right)L_2||P - P_k||_F\right].$$

Let $T_0$ be the period that the number of samples collected in the exploration phases exceeds $N_0$, that is,

$$T_0 := \inf_{t\geq 1}\{\sum_{n=1}^{t}\mathbb{1}_{(n\in H_k, \text{ for some } k)} \geq N_0\}.$$

If $T \geq T_0$ after episode $k_0$, then from Proposition 1, under the confidence level $\delta_k = \delta/k^3$, we have

$$||(\boldsymbol{\mu}_k)^m - (\boldsymbol{\mu})^m||_2 \leq C_1\sqrt{\frac{\log(\frac{6(S^2+S)k^3}{\delta})}{\tau_1 k}}, \quad m \in \mathcal{M},$$

$$||P_k - P||_2 \leq C_2\sqrt{\frac{\log(\frac{6(S^2+S)k^3}{\delta})}{\tau_1 k}}.$$

Together with the fact that the vector norm and matrix norm satisfy

$$||(\boldsymbol{\mu}_k)_i - (\boldsymbol{\mu})_i||_1 \le \sum_{m=1}^{M} ||(\boldsymbol{\mu}_k)^m - (\boldsymbol{\mu})^m||_1 \le \sum_{m=1}^{M} \sqrt{M} ||(\boldsymbol{\mu}_k)^m - (\boldsymbol{\mu})^m||_2, \quad i \in \mathcal{I},$$

$$||\boldsymbol{\mu}_k - \boldsymbol{\mu}||_1 = \max_i ||(\boldsymbol{\mu}_k)_i - (\boldsymbol{\mu})_i||_1 \le \sum_{m=1}^{M} ||(\boldsymbol{\mu}_k)^m - (\boldsymbol{\mu})^m||_1 \le \sum_{m=1}^{M} \sqrt{M} ||(\boldsymbol{\mu}_k)^m - (\boldsymbol{\mu})^m||_2,$$

$$||P - P_k||_F \le \sqrt{M} ||P - P_k||_2,$$

we obtain with probability at least $1 - \frac{5}{2}\delta$, the regret in exploitation phase can be bounded by

$$\sum_{k=1}^{K} \sum_{t \in E_k} \rho^* - \bar{c}(b_t, I_t)$$

$$\le KD + D\sqrt{2T \ln(\frac{1}{\delta})} + \sum_{k=k_0}^{K} \tau_2 \sqrt{k} \left( D + 1 + \left(1 + \frac{D(1-\alpha)}{2}\right) L_1 \right) M\sqrt{M} C_1 \sqrt{\frac{\log(\frac{6(S^2+S)k^3}{\delta})}{\tau_1 k}}$$

$$+ \sum_{k=k_0}^{K} \tau_2 \sqrt{k} \left(1 + \frac{D(1-\alpha)}{2}\right) L_2 \sqrt{M} C_2 \sqrt{\frac{\log(\frac{6(S^2+S))k^3}{\delta})}{\tau_1 k}} + T_0 \rho^*$$

$$\le KD + D\sqrt{2T \ln(\frac{1}{\delta})} + T_0 \rho^*$$

$$+ K\tau_2 \left[ \left(D + 1 + \left(1 + \frac{D(1-\alpha)}{2}\right) L_1 \right) M^{3/2} C_1 + \left(1 + \frac{D(1-\alpha)}{2}\right) L_2 M^{1/2} C_2 \right] \sqrt{\frac{\log(\frac{6(S^2+S)K^3}{\delta})}{\tau_1}}.$$

$$(32)$$

*Step 3: Summing up the regret*

Combining (22) and (32), we can get that with probability at least $1 - \frac{5}{2}\delta$, the first term of regret (17) is bounded by

$$\sum_{t=1}^{T} \rho^* - \bar{c}(b_t, I_t)$$

$$\le (K\tau_1 + T_0)\rho^* + KD + D\sqrt{2T \ln(\frac{1}{\delta})}$$

$$+ K\tau_2 \left[ \left(D + 1 + \left(1 + \frac{D(1-\alpha)}{2}\right) L_1 \right) M^{3/2} C_1 + \left(1 + \frac{D(1-\alpha)}{2}\right) L_2 M^{1/2} C_2 \right] \sqrt{\frac{\log(\frac{6(S^2+S)K^3}{\delta})}{\tau_1}}.$$

$$(33)$$

Finally, combining (18) and (33), we can see that with probability at least $1 - \frac{7}{2}\delta$, the regret presented in (17) can be bounded by

$$\mathcal{R}_T \le (K\tau_1 + T_0)\rho^* + KD + D\sqrt{2T \ln(\frac{1}{\delta})} + \sqrt{2T \ln \frac{1}{\delta}}$$

$$+ K\tau_2 \left[ \left(D + 1 + \left(1 + \frac{D(1-\alpha)}{2}\right) L_1 \right) M^{3/2} C_1 + \left(1 + \frac{D(1-\alpha)}{2}\right) L_2 M^{1/2} C_2 \right] \sqrt{\frac{\log(\frac{6(S^2+S)K^3}{\delta})}{\tau_1}}).$$

Note that

$$\sum_{k=1}^{K-1} \tau_1 + \tau_2 \sqrt{k} \le T \le \sum_{k=1}^{K} \tau_1 + \tau_2 \sqrt{k},$$

so the number of episodes $K$ is bounded by $(\frac{T}{\tau_1+\tau_2})^{2/3} \le K \le 3(\frac{T}{\tau_2})^{2/3}$.

Thus, we have shown that with probability at least $1 - \frac{7}{2}\delta$,

$$\mathcal{R}_T \le CT^{2/3}\sqrt{\log\left(\frac{9(S+1)}{\delta}T\right)} + T_0\rho^*,$$

where $S = 2I$, and

$$C = 3\sqrt{2}\left[\left(D + 1 + \left(1 + \frac{D(1-\alpha)}{2}\right)L_1\right)M^{3/2}C_1 + \left(1 + \frac{D(1-\alpha)}{2}\right)L_2M^{1/2}C_2\right]\tau_2^{1/3}\tau_1^{-1/2}$$

$$+ 3\tau_2^{-2/3}(\tau_1\rho^* + D) + (D+1)\sqrt{2\ln(\frac{1}{\delta})}.$$

Here, $M$ is the number of Markov chain states, where $L_1 = \frac{4M(1-\epsilon)^2}{\epsilon^2\min\{\mu_{\min}, 1-\mu_{\max}\}}$, $L_2 = \frac{4M(1-\epsilon)^2}{\epsilon^3} + \sqrt{M}$, with $\epsilon = \min_{1\le i,j\le M} P_{i,j}$, $\alpha = 1 - \frac{\epsilon}{1-\epsilon}$, and $\mu_{\max}$, $\mu_{\min}$ are the maximum and minimum element of the matrix $\mu$ respectively. The proof of Theorem 1 is therefore complete.

## E.1 Proof of Proposition 4

*Proof.* For each episode $k = 1, 2, \cdots, K$, let $E_k$ be the index set of the $k$th exploration phase, and $E = \cup_{k=1}^K E_k$ be the set of all exploitation periods among the horizon $T$ and $v_k$ be the value function of the optimistic POMDP at the $k$th exploitation phase. For an arbitrary time $t$, let $n = \sum_{i=1}^t \mathbb{1}_{i \in E}$, which means the number of exploitation periods up to time $t$. Define a stochastic process $\{Z_n, n \ge 0\}$:

$$Z_0 = 0,$$

$$Z_n = \sum_{j=1}^n \mathbb{E}^\pi[v_{k_j}(b_{t_j+1}^{k_j})|\mathcal{F}_{t_j}] - v_{k_j}(b_{t_j+1}^{k_j}),$$

where $k_j = \{k : j \in E_k\}$ and $t_j = \min\{t : \sum_{i=1}^t \mathbb{1}_{i \in E} = j\}$ mean the corresponding episode and period of $j$th exploitation, respectively.

We first show that $\{Z_n, n \ge 0\}$ is a martingale. Note that $\mathbb{E}^\pi[|Z_n|] \le \sum_{j=1}^n \text{span}(v_{k_j}) \le nD < TD < \infty$. It remains to show $\mathbb{E}^\pi[Z_n|\mathcal{F}_{n-1}] = Z_{n-1}$ holds, i.e., $\mathbb{E}^\pi[Z_n - Z_{n-1}|\mathcal{F}_{n-1}] = 0$. Note that

$$\mathbb{E}^\pi[Z_n - Z_{n-1}|\mathcal{F}_{n-1}] = \mathbb{E}^\pi[\mathbb{E}^\pi[v_{k_n}(b_{t_n+1}^{k_n})|\mathcal{F}_{t_n}] - v_{k_n}(b_{t_n+1}^{k_n})|\mathcal{F}_{n-1}] = 0,$$

where the last equality is due to $n - 1 \le n \le t_n$ then applying the tower property.

Therefore, $\{Z_n, n \ge 0\}$ is a martingale for any given policy $\pi$. Moreover, by Proposition 2, we have

$$|Z_n - Z_{n-1}| = |\mathbb{E}^\pi[v_{k_n}(b_{t_n+1}^{k_n})|\mathcal{F}_{t_n}] - v_{k_n}(b_{t_n+1}^{k_n})| \le \text{span}(v_{k_n}) \le D.$$

Thus, $\{Z_n, n \ge 0\}$ is a martingale with bounded difference.

Let $\bar{N} = \sum_{i=1}^T \mathbb{1}_{i \in E_k}$ and apply the Azuma-Hoeffding inequality [9], we have

$$\mathbb{P}(Z_{\bar{N}} - Z_0 \ge \epsilon) \le \exp\left(\frac{-\epsilon^2}{2\sum_{t=1}^{\bar{N}} D^2}\right).$$

Note that $\bar{N} \le T$ and $Z_{\bar{N}} = \sum_{k=1}^K \sum_{t \in E_k} \mathbb{E}^\pi[v_k(b_{t+1})|\mathcal{F}_t] - v_k(b_{t+1})$. Thus, setting $\epsilon = D\sqrt{2T\ln(\frac{1}{\delta})}$, we can obtain

$$\mathbb{P}\left(\sum_{k=1}^K \sum_{t \in E_k} \mathbb{E}^\pi[v_k(b_{t+1})|\mathcal{F}_t] - v_k(b_{t+1}) \ge D\sqrt{2T\ln(\frac{1}{\delta})}\right) \le \delta.$$

Hence we have completed the proof. $\square$