# OpenReview forum: "Regime Switching Bandits"
_NeurIPS.cc/2021/Conference — NeurIPS 2021 Poster_

### Official Review · Reviewer_q6bE · 2021-07-08

**Rating:** 7
**Confidence:** 3

**Summary:**

The authors study a variation of the multi-armed bandit problem where the rewards are neither adversarial nor stochastic. The rewards are actually random variables drawn from distributions that change based on an unknown state.
This changing state is modeled by a finite-state Markov chain, and the learner does not know the current state, nor the matrix and reward distribution.
The difficulty of this problem seems to be controlled by two different aspects: how strong is the oracle you compare the learner to, and how much budget has the environment to change the state that it is in.

The oracle is the best action in hindsight, then results for adversarial MAB would apply, which give  $ \tilde O(\sqrt T)$ types of regret bounds. The authors consider a much stronger oracle that is aware of the transition and the reward matrix of the MDP, but not of the current underlying state.
In their problem setting, the changing budget is linear in T.

The authors present an algorithm for this problem and derive a O(T^{2/3}\sqrt{\ln{T}}) regret bound against the previously mentioned oracle.
Their algorithm combines several concepts such as the spectral method of moments methods to estimate the parameters of the MDP and the commonly used UCB method.





**Limitations And Societal Impact:**

The authors clearly mention that a current limitation of their work is the fact that the theoretical guarantees that they derive for their algorithms are not optimal.
It would be good for the lower bound section

**Main Review:**

The problem of regime-switching bandits has been studied on multiple occasions in the past, but this work appears to be the first to consider a setting where the switching budget is linear while simultaneously considering a strong oracle.
Achieving any sublinear bound for this problem is already a significant contribution to this problem.

The paper is nicely written and clear for the most part.
The proofs are available in the appendix, and they appear to be quite thorough.
They also provide experiments where their algorithm outperforms different baselines.

I find that the lower bound section (Remark 4) is not very clear. It would be useful to have a clearer idea of what the suboptimality gap is for this problem.

====== Minor comments and typos ======

line 293 exportation -> should it rather be exploration?


**Time Spent Reviewing:**

7

---

> ### Author Response · Authors · 2021-08-10
> **Responses to the comments**
>
> Thank you for your careful reading and excellent comments for our paper. We have tried our best to address your comments below.
>
> - *The lower bound section is not very clear. It would be useful to have a clearer idea of what the suboptimality gap is for this problem.*
>
>  Thank you for this question. Currently there is a gap between the upper bound of our algorithm $O(T^{2/3})$ and the lower bound for the classic MAB problems $O(T^{1/2})$. It is unclear which is the optimal regret bound for the regime switching bandits problem. We tend to believe that $O(T^{1/2})$ is the fundamental limit. This is because, as mentioned in Remark 4, our algorithm uses separate exploration/exploitation phases. In the exploration phase, arms have to be pulled purely randomly in order to estimate the parameters. Such naive exploration without any maximization may lead to unnecessary regret. In fact, it resembles the structure of the explore-then-commit algorithm (Chapter 6 of [20]) for the classic MAB problem, which turns out to have the suboptimal regret $O(T^{2/3})$ as well.
> One natural direction that may potentially improve our algorithm and close the gap is to integrate the exploration and exploitation. We may follow the design similar to UCRL2 in Jaksch et al. 2010 [25]: in each episode, we use the parameters estimated in previous episodes and use the *optimistic* policy in the confidence region. The observations are then used to update the estimation. This can lower the cost for exploration because the naive exploration is replaced by a near-optimal policy when the confidence region is small. However, it requires the spectral estimator to work with non i.i.d. samples generated by the optimistic belief-based policy. The design of the spectral estimator makes it hard to handle non i.i.d. observations with complex history dependency. We may replace the spectral estimator by other estimators with better theoretical properties. The likelihood-based estimators may be a promising candidate (Yang et al 2017), but they have not been fully extended to POMDPs yet. In summary, we need better estimators for POMDP/HMMs (which is a dynamic research area itself) with finite-sample guarantees to improve the upper bound.
>
> Yang, Fanny, Sivaraman Balakrishnan, and Martin J. Wainwright. "Statistical and computational guarantees for the Baum-Welch algorithm." The Journal of Machine Learning Research 18.1 (2017): 4528-4580.
>
> - We will also fix the typos in our revision.

---

### Official Review · Reviewer_JZ7M · 2021-07-10

**Rating:** 6
**Confidence:** 3

**Summary:**

In this paper, the authors consider a variation of the well-known MAB problem where the Multi-Arm bandit can be in different regimes modelled by a common finite-state Markov Chain unobservable to the learner.
The means of the different arms can vary in the different regimes and the learner needs to infer both the transition matrix and the reward distribution in the different states.
The contribution is threefold: a new bandit formulation, an algorithm for solving this new problem and the theoretical analysis of the proposed algorithm.

**Limitations And Societal Impact:**

I think the authors also did a good job for explaining the limitation of their work:
- the computation of the optimal average reward is hard and given the confidence sets solving the optimisation program from line 8 in Algorithm 2 can be problematic.
- they acknowledge the gap between the upper and the lower bound


**Main Review:**

The study of non-stationary bandits with unobservable Markov chain has seen little treatment and the proposed solution is a first step towards a better understanding of this problem. Compared to other existing non-stationary models, the proposed approach here has a sublinear regret even if the state is changing at every time-step. I think that the proposed formulation can be of interest for the community.

The paper is overall clearly written, the authors did a good job mentioning different related works, how this work differs from others and the main technical issues due to the POMDP structure of the problem. I enjoyed reading the paper.

While the authors clearly explain the main difficulty for the different aspects of the problem: existing tools for estimating the transition matrix requires i.i.d samples, the estimation of the means in the different states, I have the feeling that there is not much novelty in the way those issues are solved.

Most of the work consists in bringing together tools from separate works (which I agree is already a contribution) but the proposed algorithm is not surprising. By completely separating the learning aspects and the tracking aspects with distinct exploration and exploitation phase most of the theoretical difficulties are avoided, limiting the novelty of this paper.

I think that the paper can be improved with a better understanding of the gap between the lower-bound and the upper-bound. Does the proposed algorithm recover the \sqrt{T} bound if the mean is the same across the different states. The experiments section is not really convincing, 50 independent replications is not enough (few hundreds would be more convincing).
How was the sliding window of SW-UCB tuned in the experiment ?

Remarks:
I was a bit surprised with line 9 and 10 of Algorithm 2, after the exploration phase from episode $k$, one has an estimate of the transition matrix and the different means in the different states. Then the belief is updated using all previous rounds (from 1 to the current time), not only using the information from episode k. I think discussing the complexity of the algorithm because of this step would be of interest, in particular this also implies that all the rewards and choice of arms have to be stored from the beginning of the experiment which is usually not the case for bandit algorithms.

**Time Spent Reviewing:**

Around 6 hours

---

> ### Author Response · Authors · 2021-08-10
> **Responses to  the comments**
>
> Thank you for your careful reading and excellent comments for our paper. We have tried our best to address your comments below.
>
> - *The paper can be improved with a better understanding of the gap between the lower-bound and the upper-bound. Does the proposed algorithm recover the $\sqrt{T}$ bound if the mean is the same across the different states?*
>
> Currently there is a gap between the upper bound of our algorithm $O(T^{2/3})$ and the lower bound for the classic MAB problems $O(T^{1/2})$. It is unclear which is the optimal regret bound for regime switching bandits. We tend to believe that $O(T^{1/2})$ is the fundamental limit. This is because, as mentioned in Remark 4, our algorithm uses separate exploration/exploitation phases. In the exploration phase, arms have to be pulled purely randomly in order to estimate the parameters. Such naive exploration without any maximization may lead to unnecessary regret. In fact, it resembles the structure of the explore-then-commit algorithm (Chapter 6 of [20]) for the classic MAB problem, which turns out to have the suboptimal regret $O(T^{2/3})$ as well.
> With special structures, such as a common mean for all states, special-purpose algorithms can be designed to achieve the optimal rate $O(T^{1/2})$. Unfortunately, our algorithm doesn’t adapt to the special structure. In fact, the spectral estimator no longer works because of the degeneracy as it cannot differentiate between states from the observations.
> One natural direction that may potentially improve our algorithm and close the gap is to integrate the exploration and exploitation. We may follow the design similar to UCRL2 in Jaksch et al. 2010 [25]: in each episode, we use the parameters estimated in previous episodes and use the *optimistic* policy in the confidence region. The observations are then used to update the estimation. This can lower the cost for exploration because the naive exploration is replaced by a near-optimal policy when the confidence region is small. However, it requires the spectral estimator to work with non i.i.d. samples generated by the optimistic belief-based policy. The design of the spectral estimator makes it hard to handle non i.i.d. observations with complex history dependency. We may replace the spectral estimator by other estimators with better theoretical properties. The likelihood-based estimators may be a promising candidate (Yang et al 2017), but they have not been fully extended to POMDPs yet. In summary, we need better estimators for POMDP/HMMs (which is a dynamic research area itself) with finite-sample guarantees to improve the upper bound.
>
> Yang, Fanny, Sivaraman Balakrishnan, and Martin J. Wainwright. "Statistical and computational guarantees for the Baum-Welch algorithm." The Journal of Machine Learning Research 18.1 (2017): 4528-4580.
>
> - *The experiments section is not really convincing, 50 independent replications is not enough (few hundreds would be more convincing). How was the sliding window of SW-UCB tuned in the experiment ?*
>
> We have increased to 500 independent replications, and find that our algorithm still outperforms the benchmark algorithms for the examples considered.
> The length of the sliding window for SW-UCB is tuned from 100 to 2000 time steps. The best tuned window length for the example studied is 1300. We will clarify this point in the revised version of the paper.
>
> - *It is surprised that the belief is updated using all previous rounds (from 1 to the current time), not only using the information from episode $k$. Discussing the complexity of the algorithm would be of interest.*
>
> This is a great point and indeed a significant difference from other bandit algorithms. The reason that we update the belief using rewards and arms from all previous rounds is to better control the error in belief $|b_t^k – b_t|$, where $b_t$ is the belief computed under the true model and $b_t^k$ is the belief computed under the estimated model in the $k$-th episode. One can only make sure that the initial beliefs are the same $b_1^k= b_1$, which serves as the base for the error calculation. This allows us to apply Proposition 3 and simplify the regret analysis. We agree with the referee that due to this step, the rewards and choice of arms have to be stored from the beginning of the experiment, so the memory/storage requirement can grow over time. We will clarify this point in the revised version of the paper.

---

> > ### Comment · Reviewer_JZ7M · 2021-09-02
> > **Response to rebuttal.**
> >
> > I thank the authors for the detailed comments to my different remarks.
> > I think that storing all the previous choices and rewards is an important aspect of the algorithm but assuming that the authors clarify this in the revised version, in my opinion the paper is above the acceptance threshold.

---

### Official Review · Reviewer_oiQG · 2021-07-16

**Rating:** 7
**Confidence:** 4

**Summary:**

This paper studies a non-stationary and finite bandit model with Markovian regime-switching rewards, which is termed “regime switching bandit”. As mathematically formalized in the paper, in this new bandit model, the reward distributions of various arms are modulated by an underlying finite-state Markov chain, with unknown transition matrix, and whose state is assumed unobservable. The paper develops a learning algorithm called SEEU, which combines spectral estimators for the Markov chain involved with the optimism principle (together with some forced exploration). Ignoring logarithmic factors, SEEU is shown to achieve a regret of $O(T^{2/3})$, with high probability, against an oracle who knows reward distributions and the transition function of the Markov chain, but as the learner, does not observe the state of the Markov chain.

**Ethical Concerns:**

None.

**Limitations And Societal Impact:**

See above.

**Main Review:**

The paper studies a novel and interesting non-stationary bandit model, which, to the best of my knowledge, does not match those in the existing literature. Despite being interesting from a theoretical standpoint, it is also well motivated by some existing applications.

The authors do a good job of precisely formalizing the “regime switching bandit” model. They provide an adequate literature review and make a fair and informative comparison between their model and other existing non-stationary bandit models. Compared to the existing literature, the presented formulation considers a stronger oracle (if not the strongest possible), who follows the optimal POMDP policy. This is in my opinion one strong aspect of the presented formulation.

The presented SEEU algorithm follows the optimism principle combined with a forced exploration strategy, and makes use of spectral estimators to estimate unknown transition probabilities and mean rewards. Forced exploration appears to be sub-optimal in many regimes, but as it turns out, for regime switching bandits avoiding that is not that easy, due to the complicated nature of the problem.

To my best knowledge, the presented SEEU algorithm, though being rate sub-optimal, is the first algorithm amenable to the presented model. I suspect that the regret of SEEU, in terms of $T$, is unimprovable, mainly because of forced exploration, as the authors mentioned already.

Although the design of SEEU makes sense, it has some drawbacks, which include dependence on some unknown parameters, very weak regret bounds, and unimplementability --- See the detailed discussion below. I could not check all the proofs in the appendix in this limited time, but they appear correct to me.

Detailed Comments:

1- The regret under SEEU has a prohibitively large dependence on $\epsilon = P_{\min}$ as follows. The main term in the regret bound depends on $C$, which is $O(DL_1C_1M^{3/2} + DL_2C_2M^{1/2})$. Here $D = D(\epsilon)$ grows as $O((1-\alpha)^{-3})$, with $\alpha$ being proportional to $1-\epsilon$. Moreover, $L_1 = O(M\epsilon^{-2})$ and $L_2 = O(M\epsilon^{-3})$ – I already ignored other factors, such as $\mu_{\min}$, which would only worsen the bound.
All in all, the multiplicative factor $C$ scales as $O(\epsilon^{-5}M^{5/2}C_1 + \epsilon^{-6} M^{3/2}C_2) )$, where $C_1$ and $C_2$ further depend on mixing times related quantities (which could be large in some case).

In line 287, the authors report the dependence of $C$ only on $M$ and $I$, remaining silent about its prohibitive dependence on $\epsilon = P_{\min}$. Although $\epsilon$ is assumed away from zero, it could be arbitrarily small, making the regret bound essentially uninformative, if not useless.  Of course, I understand $C$ would necessarily depend on $\epsilon$, but it seems the current analysis is quite crude so that the final bound has such an unacceptable dependence. Therefore, the derived regret bound, which is already rate-sub-optimal, suffers from a crude multiplicative constant, which could come from a crude regret analysis.

2- Dependence on prior knowledge. As already remarked in the paper, to determine $C_1$ and $C_2$, as well as $\tau_1$ and $\tau_2$, SEEU relies on some parameters of the hidden Markov chain, including its mixing time and minimal stationary distribution. All these quantities are crucial for the algorithm to work properly. This is a weak aspect of SEEU, making its applicability quire restricted. The parameters needed to compute $C_1$ and $C_2$ (or upper bounds on them) are a priori unknown, and even assuming an upper bound on them is sort of a strong assumption. I appreciate the authors’ remarks in Section 3.3 in this regard, but their idea in line 260 to remedy this would not work in my opinion, because the corresponding estimation errors might be non-trivial to control.

3- Another weak aspect of SEEU is related to solving the optimistic POMDP. Assuming access to an oracle for solving (2) might be tolerated as the main goal here is to achieve statistical efficiency. However, the optimistic MDP in SEEU is totally non-trivial to solve, even assuming access to the said oracle. This further renders SEEU unimplementable. Although discretization of $\mathcal C_k$ could be an option, the corresponding discretization error could impact the regret in a non-trivial manner. In my view, the algorithm needs further elaboration in this regard.

Despite introducing a novel and interesting bandit model, the presented SEEU algorithm and its analysis, which constitute the main contributions here, are not practical or significant enough for several reasons motivated above. I therefore believe the paper could still benefit from further elaboration, and in its current status, it does not pass the acceptance threshold by a good margin. All in all, I vote for a score of 6, mainly scoring the paper due a new interesting bandit model. However, I would like to hear the authors’ responses to my raised comments.

A minor comment:

Line 298: the total number of episodes … satisfy => … satisfies


**Time Spent Reviewing:**

5 hours

---

> ### Author Response · Authors · 2021-08-10
> **Responses to the comments**
>
> Thank you for your careful reading and excellent comments for our paper. We have tried our best to address your comments below.
>
> - *The large dependence on $\epsilon$.*
>
> Thank you for this point. Following your comment, we have tried alternative techniques in the revision and we are able to improve the dependency on $\epsilon$ from $\epsilon^{-6}$ to $\epsilon^{-5}$ by using a refined regret analysis. First, the current upper bound in Proposition 3 is uniform in time, and it is crude for small $t$ since the initial belief is the same with $b_1 = \hat b_1.$ One can improve the bound by multiplying it with a factor of $1 - \alpha^{t-1}$ where $\alpha \approx 1-\epsilon$ is given in Proposition 2. This follows from Proposition 3 of De Castro et al. 2017 [18] and its proof.  Using this observation, in the proof of regret bound, we can multiply $1-\alpha$ to the term in Equation (28) (Line 675 - 676) in the appendix, since here we consider the one-step update of the belief under two different sets of model parameters. Hence, the quantities $D L_1$ and $D L_2$ in the constant C of the regret bound can be multiplied by $1-\alpha$, which is of order $\epsilon$. Therefore, the multiplicative factor $C$ scales as $O(\epsilon^{-4} M^{5/2} C_1 + \epsilon^{-5} M^{3/2} C_2 )$, which improves our previous result in terms of the $\epsilon$ dependency.
>
> - *Dependence on prior knowledge. As already remarked in the paper, to determine C1 and C2, as well as τ1 and τ2, SEEU relies on some parameters of the hidden Markov chain, including its mixing time and minimal stationary distribution.*
>
> Below we provide some basic ideas for a subroutine to estimate the constants $C_i$ and the impact on the regret bound. The details have to be worked out rigorously in the revision.
> We can set up a preliminary phase at the beginning of the horizon and pull arms purely randomly. The preliminary phase is of length $O(T^{2/3})$ and thus the regret incurred in the phase is $O(T^{2/3})$, not worsening the total regret. We apply the spectral estimator to the observations from the preliminary phase. The confidence bound would look like $||\mu- \hat{\mu}||_2 \leq C_1T^{-1/3}\sqrt{\log(6\frac{S^2+S}{\delta})}$ and $||P - \hat{P}||_2\leq C_2T^{-1/3}\sqrt{\log(6\frac{S^2+S}{\delta})}$ by Proposition 1. Note that we still do not know $C_1$ and $C_2$ yet. Because $C_1$ and $C_2$ are determined by $\mu$ and $P$, for any given $P$ and $\mu$, we can calculate $C_1$ and $C_2$ and check whether the given $P$ and $\mu$ fall into the confidence bound above. This gives us a set of $(P,\mu)$, and thus $(C_1, C_2)$, that satisfy the confidence bounds of Proposition 1. This can be regarded as the confidence bound of $C_1$ and $C_2$. The confidence level can be set to $\delta=O(T^{-1/3})$. Therefore, when we choose the upper bound of $C_1$ and $C_2$ in the confidence bounds, they are indeed the upper bounds with probability no less than $1-O(T^{-1/3})$. The total regret due to the error is at most $T\times O(T^{-1/3}) = O(T^{2/3})$. This shows that the preliminary phase doesn’t change the total regret rate.
> The choices of $\tau_1$ and $\tau_2$ do not rely on the unknown parameters of the hidden markov chain. They can be tuned as hyperparameters. The requirement of a large $\tau_1$ in our current theorem statement is not essential and can be removed. As long as $\tau_1$ is greater than the number of arms,  the regret bound will still hold by replacing $T_0$ in Theorem 1 with another constant that is independent of T. See the responses to Reviewer pMqb for further details.
>
> - *Another weak aspect of SEEU is related to solving the optimistic POMDP. Although discretization of $\mathcal{C}_k$ could be an option, the corresponding discretization error could impact the regret in a non-trivial manner. In my view, the algorithm needs further elaboration in this regard.*
>
> This is a great point. When implementing the algorithm, the discretization error will affect the regret, although it can be controlled arbitrarily well with sufficient computational capacity. Below we characterize its impact on regret. The discretization kicks in when we want to find the optimistic POMDP in the confidence region, whose gain is denoted as $\rho^k$ in episode $k$. In practice, we have to discretize the confidence region into grid points and find the optimistic POMDP among the finite set. Suppose one can obtain an approximate optimistic model with error $\epsilon_k$, that is, suppose we can find a model with the gain $\tilde \rho_k \ge \rho^k - \epsilon_k$. Then we can analyze the impact of the discretization error on the regret as follows. We can infer from Line 637 - 640 in the appendix that the extra regret incurred due to the discretization error is given by $\sum_{k=1} ^K \sum_{t \in E_k} \epsilon_k =  \tau_2 \sum_{k=1} ^K  \sqrt{k} \epsilon_k,$ where $E_k$ denotes the exploitation phase in episode $k$. Note that the order of $K$ is $T^{2/3}$. Hence if the discretization error can be controlled at $\epsilon_k = c/\sqrt{k}$, then the extra regret is simply $c \tau_2 K$, which is $O(T^{2/3})$. On the other hand, if the discretization error $\epsilon_k$ is a constant $c>0$ in all the exploitation phases, then the extra regret incurred is of the order $\sum_{k=1} ^K  \sqrt{k} c \approx c \cdot K^{3/2}$, which is of order $ c T$. There is a trade-off between the additional computational complexity due to discretization and the regret bound. For instance, to control the discretization error at $\epsilon_k = c/\sqrt{k}$, the computational cost is higher compared with the case $\epsilon_k = c$. In general, it remains open to find efficient methods to solve the optimistic POMDP approximately in the high-dimensional setting. We will add these discussions to the revision.
>
> - We will also fix the typos in our revision.

---

> > ### Comment · Reviewer_oiQG · 2021-09-01
> > **Response to Rebuttal**
> >
> > I would like to thank the authors for their efforts to answer my questions and address my comments. I have read all the reviews as well as the rebuttal, and I confirm that the rebuttal addresses some of my major comments. Assuming that the authors implement the promised changes, I increase my score to 7, thus recommending acceptance.

---

### Official Review · Reviewer_pMqb · 2021-07-22

**Rating:** 6
**Confidence:** 3

**Summary:**

This paper studies a multi-armed bandit problem where the reward of the arm depends on the hidden state. The state changes according to the dynamics of the MDP, which means that the arms chosen in the past will influence the states chosen in the future time steps. The number of states and available actions at each state in the MDP is assumed to be finite, and the reward is assumed to be drawn from a distribution with finite support. This work studies the setting where the learner the transition matrix and the reward distribution of the arm for a given state and also the learner doesn't observe the reward. The main contribution of the paper is the algorithm that is shown to achieve T^2/3 regret in the described setting, where the regret is measured against the best policy that doesn't observe the current state and is not restricted to be a memoryless policy.

**Limitations And Societal Impact:**

There is no potential negative societal impact.

**Main Review:**

Overall, the work shares many similarities with Azizzadenesheli et al 2016, with the main difference that there the regret was measured with respect to the memoryless policy, which is a weaker result since for POMDP the optimal policy doesn't have to be memoryless. This work claims that this difficulty has been overcome and this is the first result that achieves the sublinear regret in this setting.

Concerns:

-The definition of \rho* in line 182 needs more care.

- In the paragraph between lines 192 and 202, there is no mention of the result of Azizzadenesheli et al 2016, which contradicts the state that the spectral estimation technique wasn't applied for the decision making.

- The Theorem 1 states that \tau_1 has to be large enough. How this quantity depends on other parameters of the problem? Nothing is said about \tau_2 in the statement.

- It is worth mentioning the result of Madani 1998, which has shown that finding the optimal policy is uncomputable for infinite horizon regret minimization. You also mention the lower bound in line 354, could you please provide the reference to it?

- Looks like there is no need in Theorem 3, results of Theorem 1 and 3 can be stated as one result.

**Time Spent Reviewing:**

8

---

> ### Author Response · Authors · 2021-08-10
> **Responses to the comments**
>
> Thank you for your careful reading and excellent comments for our paper. We have tried our best to address your comments below.
>
> - *The definition of $\rho^\*$ in line 182 needs more care.*
>
> We will clarify the definition of $\rho^*$ in line 182.
>
> - *There is no mention of the result of Azizzadenesheli et al 2016.*
>
> Sorry for the confusion. In the paragraph between lines 192 and 202, we are referring to the fact that for hidden markov models, there is no decision making involved. We will change the wordings to make this clear.
>
> Technical challenges compared to Azizzadenesheli et al 2016: The algorithm in Azizzadensheli et al 2016 will generally suffer linear regret in our problem setting. This is because we consider a stronger oracle which is the optimal POMDP policy, whereas they consider the optimal memoryless policy.  It is known that memoryless policies are in general not optimal and hence the gap between their oracle and our oracle can be linear. As a result, we need to design a new learning algorithm to achieve sublinear regret with our oracle. By considering the belief-based policies, several new difficulties (which are not present in Azizzadensheli et al 2016) arise in our setting. For instance, the belief states cannot be observed and we need to control its estimation error. Also, the spectral method can not be applied to samples generated from belief-based policies due to the complex history dependency; Finally, we need to develop a new approach to bound the bias span for the optimistic belief MDP in the regret analysis.
>
> - *Theorem 1 states that $\tau_1$ has to be large enough. How this quantity depends on other parameters of the problem? Nothing is said about $\tau_2$ in the statement.*
>
> A large $\tau_1$ makes Proposition 1 applicable from the first episode and thus simplifies the presentation. Theoretically, this requirement is not essential and can be removed. In fact, as long as $\tau_1$ is greater than the number of arms,  the regret bound still holds by replacing $T_0$ in Theorem 1 with another constant (larger than $T_0$) that is independent of $T$. The argument proceeds as follows: without a large $\tau_1$, we may need a burn-in period for total number of exploratory samples to exceed $N_0$ in Proposition 1 and the estimated transition probability matrix to have a minimal element exceeding $\epsilon/2$ (so that Assumption 3 is satisfied with $\epsilon$ replaced by $\epsilon/2$). The length of the burn-in period, say $S_0$, is independent of $T$ and thus doesn’t affect the regret bound significantly. After $S_0$ time steps, the confidence bounds in Proposition 1 and the uniform bound on the bias span in Proposition 2 can both be applied in the proof of the regret bound. Note that $\tau_2$ can be an arbitrary positive integer in the regret analysis. We will clarify this point in the revised version of the paper.
>
> - *It is worth mentioning the result of Madani 1998.*
>
> We will mention the result of Madani 1998 as suggested by the referee. The lower bound we refer to is the lower bound for the classical multi-armed bandit problems; See Remark 4.
>
> - *Looks like there is no need in Theorem 3, results of Theorem 1 and 3 can be stated as one result.*
>
> Theorem 1 is about the high-probability regret bound and Theorem 3 is about the expected regret bound. We felt they may apply to different settings. We agree with you that they can be combined and stated as one result as you suggested.

---

> > ### Comment · Reviewer_pMqb · 2021-09-02
> > **Comments**
> >
> > Thank you for the illuminating answers to my questions. I increased my score.

---

### Decision · Program_Chairs · 2021-09-27

**Decision:**

Accept (Poster)

**Comment:**

The reviewers agreed that the paper provides a nice contribution by presenting a learning algorithm for the problem which competitive with a much stronger baseline than what was previously used in the literature. On the other hand, several issues were raised, most importantly that (i) the algorithm design with the forced exploration part seems suboptimal and the induced separation makes combining existing results relatively easy; and (2) the algorithm can't be implemented in an efficient way yet supposedly does not achieve the optimal rate.

Nevertheless the reviewers felt that the positives outweigh the negatives, hence I recommend acceptance of the paper. In the revised version the authors should address the several problems discussed in the review process, and provide a better and more transparent comparison to Azizzadenesheli et al. (2016) (how the algorithm relates to their method), as well as including new baselines in the experiments, such as a modified version of the algorithm of  Azizzadenesheli et al., as well as some oracle baselines which allow the experimental analysis of different parts of the algorithm.